

# The mutually antagonistic effect of drought and sand burial enables the biocrust moss *Bryum argenteum* Hedw. to survive the two co-occurring stressors in an arid sandy desert

Rongliang Jia, Yun Zhao, Yanhong Gao, Rong Hui, Haotian Yang, Zenru Wang

Shapotou Desert Research and Experiment Station, Northwest Institute of Eco-Environment and Resources,

5   Chinese Academy of Sciences, 320 Donggang West Road, Lanzhou 730000, China.

*Correspondence to*: Rongliang Jia (rongliangjia@163.com)

**Abstract.** Biocrust moss is an essential soil surface bio-cover. It represents the highest succession stage among

the diverse range of surface-dwelling cryptogams (e.g., cyanobacteria, green algae, and lichen, which are also

10   referred to as biological soil crusts) and makes a major contribution to soil stability and fertility throughout arid

desert ecosystems. The soil surface represents a small ecological niche that is poikilohydric in nature. Biocrust

moss is therefore highly susceptible to drought and sand burial, which are two ubiquitous stressors in arid sandy

deserts. However, little information is available regarding the mechanism by which biocrust moss can survive

and flourish in these habitats when stressed simultaneously by the two stressors. The combined effects of

15   drought and sand burial were evaluated in a field experiment using the predominant biocrust moss, *Bryum

argenteum* Hedw., in the Tengger Desert, China. Drought was simulated by applying distilled water in three

artificial rainfall regimes at 8-day intervals in spring and autumn: 4 and 6 mm (average rainfall, control), 2 and

3 mm (double drought), and 1 and 1.5 mm (fourfold drought), respectively. The effect of sand burial was

determined by applying six treatments, i.e., sand depths of 0 (control), 0.5, 1, 2, 4, and 10 mm. The four

20   parameters of chlorophyll a content, PSII photochemical efficiency, regeneration potential, and shoot upgrowth

were evaluated in the moss. It was found that the combined effects of drought and sand burial did not

exacerbate the single negative effects of the four parameters tested. Drought significantly ameliorated the

negative effects of deep sand burial on the retention of chlorophyll a content, PSII photochemical efficiency,

and regeneration potential of *B. argenteum*. Sand burial diminished and even reversed the negative effects of

drought on the maintenance of chlorophyll a content, PSII photochemical efficiency, and regeneration potential. Although drought and sand burial imposed an additive negative effect on shoot upgrowth, which suggested a trade-off between growth ability and stress tolerance, their mutually antagonistic effect on the physiological vigor of *B. argenteum* provided an opportunity for the biocrust moss to overcome the two co-occurring stressors. In addition to providing a strong stress tolerance, drought and sand burial may provide an important mechanism for the biodiversity maintenance of biocrust mosses in arid sandy ecosystems.

## 1 Introduction

Drought is the most common stressor constraining biological activity in dryland ecosystems (Whitford, 2002; Huxman et al., 2004). The predicted increases in the frequency and severity of droughts is likely to generate more profound consequences for community structure and ecosystem functioning in arid and semiarid ecosystems (IPCC, 2007; Smith, 2011; Weber et al., 2016). In arid sandy ecosystems, drought generally occurs alongside another ubiquitous disturbance, sand burial, due to the lowering of the threshold friction velocities of the upper soil surface (Belnap and Gillette, 1998; Li, 2012). Sand burial can alter various physical factors such as moisture, temperature, aeration, and other aspects of the plant and soil micro-environment. It can therefore act as a filter eliminating sensitive species, and it plays a significant role in determining the composition and distribution of desert vegetation (Maun, 1998, 2008). Therefore, in habitats stressed simultaneously by drought and sand burial (e.g., arid desert ecosystems) throughout China and worldwide, the growth and distribution of plants is expected to be limited. This is evidenced by mobile sand dunes, with negligible vegetation cover as an extreme example.

Biocrust moss is an essential soil surface bio-cover. It represents the highest succession stage among the diverse range of surface-dwelling cryptogams (e.g., cyanobacteria, green algae, and lichen, which are also referred to as biological soil crusts) and makes a major contribution to soil stability and fertility throughout arid and semiarid desert ecosystems (Weber et al., 2016). The colonization and development of moss on the surface of a sand dune is an important biomarker denoting the ecosystem as being stable and healthy (Zhang et al.,




2010). Thus, the assessment, protection, and utilization of moss is a major management priority in desert

regions (Stark et al., 2004; Barker et al., 2005, Xu et al., 2009; Doherty et al., 2015).

Since the 1950s, large-scale construction and land restoration has occurred throughout the arid and

semiarid sandy areas of north China, with the aims of inhibiting the harmful effects of mobile sand movement

and recovering degraded ecosystems. One striking success has been the Shapotou revegetation system, which

was constructed to alleviate burial stress using combined applications of wind barriers, straw checkerboards,

and planting anti-drought shrubs without irrigation (as shown in Fig. 1). Sixty years later, along with the

succession of vegetation, biocrust biota have gradually colonized the area and now thrive on the previously bare

soil surface, where they constitute more than 90% of the living ground cover. As a pioneer moss species, *Bryum*

*argenteum* Hedw. dominates the soil surface in this system, with a coverage exceeding 70% and making a

major contribution to soil stability and fertility. Its role is particularly important in areas where the sand-binding

role of previously planted shrubs has weakened with time (Li et al., 2004). This phenomenon is evident

throughout other sandy desert ecosystems restored by similar methods in China, where *B. argenteum* usually

appears as the pioneer moss species. Consequently, it needs to be understood why *B. argenteum* can survive and

thrive in ecosystems stressed by both drought and sand burial, enabling it to be the pioneer species.

Species that are poikilohydric in nature lack vascular support tissues, but usually protrude above the soil

surface to receive light for photosynthesis. Also, they are completely immobile, which prevents them from

finding refuge to avoid drought stresses (Garcia-Pichel and Belnap, 1996). *Bryum argenteum* responds

negatively to drought, despite its high desiccation tolerance (e.g., Li et al., 2014). Because it grows on the

surface, *B. argenteum* is inevitably exposed to repeated sand burial of various depths. Due to its limited height

above the ground surface (1 to 25 mm) the moss can be completely submerged even when the burial depth is

shallow (Jia et al., 2008). This has generated multiple organic horizons of ''fossilized mosses'' in areas where it

has survived burial stress and barren spaces where it has not (Jia et al., 2008). Therefore, there must be a

mechanism for *B. argenteum* to adapt to and survive this combination of stressors, although it remains poorly

understood. A clear understanding of the mechanism enabling *B. argenteum* to survive the co-occurring drought

and sand burial stressors in desert areas would help to explain the distribution mechanisms of this common

species. It would also enable us to predict the consequences of climate change and to formulate management

policies and restoration practices using biocrust moss to stabilize and rehabilitate degraded flowing sandy

dunes.

Previous studies have principally focused on the individual effects of drought (Stark et al., 2004; Barker et

al., 2005; Xu et al., 2009) and sand burial (Jia et al., 2008) as stressors on desert biocrust mosses, with little

emphasis on their combination and even less on their interaction. Considering the different and even contrasting

effects of drought and sand burial on physiology and growth, it is of interest to determine if a combination of

drought and sand burial imposes a mutually antagonistic effect on the physiology and growth of *B. argenteum*,

enabling it to survive the two co-occurring stressors. Drought is reported to protect moss from heat shock (Xu

et al., 2009) and ultraviolet-B induced damage (Turnbull et al., 2009), while sand burial has been reported to

slow water loss from moss crusts (Meng et al., 2011). Therefore, our initial hypothesis is that the combination

of drought and sand burial has a mutually antagonistic effect on the physiology, regeneration, and growth of *B.*

*argenteum*. To test this hypothesis, multiple assessments of the single and combined effects of drought and sand

burial stresses were made, including measurements of the chlorophyll a content, PSII fluorescence,

regeneration potential, and growth rate.

## 2 Materials and methods

### 2.1 Study site

The study area was located in the southeastern fringe of the Tengger Desert (37° 28′ N, 105° 00′ E,

elevation 1,339 m). It lies within the transitional zone from desert steppe to steppified desert and also represents

a transitional belt between desert and oasis. The mean annual temperature is 10.6 ℃, with the minimum

temperature being -25.1 ℃ in January and the maximum being 38.1 ℃ in July. The mean annual pan potential

evaporation is around 3,000 mm, while the mean annual precipitation is 180.2 mm, more than half of which

falls in summer (June-August). The other three seasons typically experience more drought periods. The



landscape of the study region consisted of large and dense reticulate barchan chains of sand dunes, where the predominant native plants were *Hedysarum scoparium* Fisch. and *Agriophyllum squarrosum* Moq. that together covered about 1% of the ground surface. No biocrust was found on the surface of the mobile sand dunes (Li, 2012).

A no-irrigation vegetation system was established in 1956 to protect Baotou-Lanzhou railway line from sand burial. It consisted of straw-checkerboards as sand barriers to fix shifting dunes, with the subsequent planting of xerophytic shrub seedlings (*Caragana korshinskii*, *Artemisia ordosica*, *Calligonum arborescens*, etc.). The system was further expanded in 1964, 1981, and 1987. These vegetated areas were distributed parallel to the railway line, with a length of 16 km and a width of 1–2 km (Li et al., 2004). The initial shrub vegetation was gradually replaced by communities dominated by herbaceous plants due to the decreasing soil water content in the upper soil layers (Li et al., 2004). Biocrust biota then colonized and developed on the stabilized dunes, which resulted in the surface becoming increasingly stabilized. As a pioneer species, *B. argenteum* successfully colonized the revegetated area and became widespread, with a coverage exceeding 70% on windward slopes and low lying sand dunes. Although there is a gradual reduction in sand burial stress on the growth of *B. argenteum* as the biocrust moss becomes established, it is inevitably exposed to repeated wind-blown sand events, leading to dust burial of various depths. This burial is typically caused by two different processes that are seasonal in their severity. In spring, when the wind speed is usually the highest and drought is most severe (precipitation is the lowest), burial by wind-blown sand predominates. In summer and autumn, when the drought is slightly alleviated by higher levels of precipitation, animal activity (burrowing by ants, lizards, and rabbits) becomes important.

### 2.2 Sampling and treatments

Samples of intact moss crusts (85 cm$^2$, 10-cm thick), with 100% coverage of *B. argenteum* were randomly collected using cylindrical PVC dishes (104-mm diameter, 12-cm height). At the base of each dish there was a drainage outlet that was covered with strips of nylon mesh to allow excess water to be removed, while

125  preventing the loss of sand. All samples were collected from the interspaces between shrubs in the revegetated

area that was established in 1981. Sampling was conducted in late February and late August, which was about

10 days before the experiments began in spring and autumn, respectively. Samples were gently processed and

sprayed with distilled water to ensure they were moist and that the sample structure remained intact. All

samples were placed below the ground surface, with the top 2 cm left aboveground. Rain shelters were then

placed at a height of 2 m above the samples. The soil surfaces surrounding the samples were paved with a straw

curtain, which extended for 5 m beyond the shade of the shelters to prevent disturbance from sand particles

outside the study area.

Drought and sand burial stress treatments were conducted from 10 March (spring) and 1 September (autumn),

respectively. A total of 108 samples were collected for each experiment in the two different seasons and were

randomly divided into three water supply groups by applying distilled water in three artificial rainfall regimes

at 8-day intervals in spring and autumn: 4 and 6 mm (average rainfall, control), 2 and 3 mm (double drought),

and 1 and 1.5 mm (fourfold drought), respectively. To determine the effect of burial, six treatments were

applied, with depths of 0 (control), 0.5, 1, 2, 4, and 10 mm, equivalent to 0, 4.25, 8.5, 17, 34, and 85 ml of dried

sand, respectively. The sand was distributed gently and evenly over crusts that had been subjected to each of the

water-supply subgroups described above.

There were six replicates of the drought × sand burial treatment. The prescribed sand burial depths and

quantities of water applied were selected based on actual sand burial depths and precipitation levels observed

during the period of 1990-2010 (Li et al., 2012) in the study area. The duration of each experiment was 72 d.

**2.3 Measurements of the chlorophyll a content, PSII photochemical efficiency, regeneration potential,
and maximal shoot upgrowth**

On the day after each experiment was completed, the sand particles deposited over the moss were gently blown

off and the same weight of sand applied prior to the burial treatment was collected. The upper 26 mm (i.e.,

including the active moss rhizoids) and inner core (5-cm diameter) were excavated from each original sample





and placed into cylindrical plastic dishes (5-cm diameter, 28-mm height). Each dish had a drainage outlet at the bottom that was covered with a strip of nylon mesh to allow excess moisture to be removed. These smaller samples were more representative than the original samples because the edge effect of the PVC tube was removed.

The six small samples from each treatment were randomly divided into two subgroups, one for the

determination of maximal shoot upgrowth, PSII photochemical efficiency, and regeneration potential, and the other for the measurement of the chlorophyll a content of *B. argenteum*. The methods used to measure the maximal shoot upgrowth, PSII photochemical efficiency, regeneration potential, and chlorophyll a content of samples were adopted from Jia et al. (2012).

The maximal shoot elongation of *B. argenteum* was determined by the difference between the vertical

distances from the upper edge of the PVC container to the uppermost part of the crust surface prior to sand burial and after removal from the sand at the end of **t**he experiment using a Vernier caliper.

The samples were watered to saturation level and then cultured in a growth chamber (Thermoline Scientific Equipment Pty. Ltd, NSW, Australia). The photon flux density (PFD), air temperature (Ta), relative air humidity (RH), and $CO_2$ concentration (Ca) was set to 1,000 mmol m$^{-2}$ s$^{-1}$, 25$^{0}$C, 55%, and 390 mmol m$^{-2}$ s$^{-1}$, respectively,

during the day (0800-1900 h), and 0 mmol m$^{-2}$ s$^{-1}$, 15$^{0}$C, 65%, and 400 mmol m$^{-2}$ s$^{-1}$, respectively, at night (1900-0800 h the next day). The position of each sample was randomly changed every day. After a 3-d pre-acclimation, the samples were wetted again to saturation level and PSII photochemical efficiency was measured 4 h later (when the maximum value occurred). The PSII photochemical efficiency (Fv/Fm) was determined by an analysis of the slow kinetics of chlorophyll fluorescence using a PAM-2000 fluorometer (Walz,

Effeltrich, Germany). The device was adjusted to maintain a distance of 1.20 cm between the fiber optics exit plane and sample. Prior to measurement, the samples were dark adapted for 5 min and then supplemented with a sequence of weak irradiance and saturation pulses (5,000 mmol m$^{-2}$ s$^{-1}$).

These samples were then hydrated, washed, and shaken to remove any sand attached. Then, 1 mm-long portions of the upper shoots containing stem apices were isolated from each of the smaller samples and placed

on native sand that had previously been sieved and dried as described by Stark et al. (2004). The same cylindrical plastic dishes as in the previous experiment were used, and the experimental conditions were also the same, except that moisture was supplied daily. The protonemal area was determined according to the protocols described by Barker et al. (2005) after a 58-d inoculation. The measurement of the regeneration potential of detached shoots was only conducted in the spring experiment, due to some of the autumn experiment samples being broken during transportation.

The chlorophyll a content was determined on a mg g$^{-1}$ dry weight basis by high performance liquid chromatography (HPLC) using a method described by Gilmore and Yamamoto (1991). Briefly, 50 mg of dry, soil-free shoots were collected from each sample, ground, extracted in 80% acetone, and then centrifuged at 10,000 g for 5 min. After removal of the supernatant, the remaining pellet was resuspended in 100% acetone and centrifuged again at 10,000 g. The supernatants were then mixed and passed through a 20-mm filter prior to injection into a Spherisorb ODS 1 column (Alltech Associates Inc., Deerfield, IL, USA) at a flow rate of 1 cm.

### 2.4 Statistics

A one-way analysis of variance (ANOVA) was used to test for any significant differences in the data using the SPSS 21 software (SPSS Inc., Chicago, USA). Differences between the individual parameters were evaluated by least significant difference (LSD) post-hoc multiple comparisons at the 95% confidence level.

Linear regressions were performed in the ORIGIN 8.5 (OriginLab, Northampton, USA) to depict any trends in the chlorophyll a content, PSII photochemical efficiency, regeneration potential of detached moss shoots, and maximal shoot elongation to increases in drought severity /burial depth.

A detrended correspondence analysis (DCA) of the chlorophyll a content, PSII photochemical efficiency, regeneration potential (protonemal area), and shoot upgrowth of *B. argenteum* was used to determine whether linear or unimodal ordination methods should be applied. We then performed a redundancy analysis (RDA) to determine the relationships between the parameters listed above and environmental parameters. A Monte Carlo permutation test (n = 499) was used to determine the significance of all canonical axes. Both DCA and RDA

were performed using Canoco for windows 5.0 (Ithaca, NY, USA).

## 3 Results

### 3.1 Seasonal changes in *B. argenteum* cover and its response to sand burial depth

*Bryum argenteum* can completely cover a soil surface, and there was no significant difference between spring

and autumn cover when sand burial was absent (Table 1). Sand burial significantly reduced the *B. argenteum* cover in both seasons, with the decrease in autumn being significantly lower than that in spring (Table 1).

### 3.2 Interactive effects of sand burial and drought on the chlorophyll a content of *B. argenteum*

The chlorophyll a content of *B. argenteum* was generally lower in spring than under the same treatment in

autumn, with the same trend found in the response to drought, sand burial, and their combination (Fig. 2). Drought uniformly imposed negative effects on the chlorophyll a content, whereas burial by sand had a dual effect on the chlorophyll a content. The chlorophyll a content increased in treatments when the burial depth was shallow (< 2 mm) and decreased when the depth was larger (sand burial depth $\geqslant$ 2 mm).

A significant interactive effect between drought and sand burial on the chlorophyll a content of *argenteum*

was found. Drought strengthened the positive effects of shallow burial and mediated the negative effects of deep burial with regard to chlorophyll a retention (Fig. 2c, d). In addition, sand burial weakened and even reversed the negative effects of drought on the retention of the chlorophyll a content in *B. argenteum* (Fig. 2a, b).

### 3.3 Interactive effects of sand burial and drought on the PSII photochemical efficiency of *B. argenteum*


The PSII photochemical efficiency of *B. argenteum* displayed the same trends as the chlorophyll a content in response to drought, sand burial, and their combination, although it was generally lower in spring than under the same treatment in autumn (Fig. 3). Drought consistently exerted negative effects on the PSII photochemical efficiency, while burial by sand had a dual effect on the PSII photochemical efficiency. The PSII photochemical

efficiency increased in treatments where the burial depth was shallow (< 2 mm) and decreased when the depth

was larger (sand burial depth ≥ 2 mm).

A dramatic interactive effect between sand burial and drought on the PSII photochemical efficiency of *B. argenteum* was observed. Drought strengthened the positive effects of shallow burial and ameliorated the negative effects of deep burial with regard to PSII photochemical efficiency (Fig. 3c, d). Sand burial diminished

and even reversed the negative effects of drought on the retention of the PSII photochemical efficiency (Fig. 3a, b).

**3.4 Interactive effects of sand burial and drought on the regeneration potential of detached shoots of *B. argenteum***

Drought imposed negative effects on the regeneration potential of detached shoots of *B. argenteum* (Fig. 4a, b), while burial by sand had a dual effect on the regeneration potential. The regeneration potential increased in treatments where the burial depth was shallow (< 2 mm) and to decrease when the depth was larger (sand burial depth ≥ 2 mm).

There was a remarkable interactive effect of sand burial and drought on the regeneration potential of *B.*

*argenteum*. Sand burial alleviated and even converted the negative effects of drought into positive effects with regard to the regeneration potential of detached shoots. Drought enhanced the positive effects of shallow burial and eased the negative effects of deep burial on the regeneration potential of detached shoots.

**3.5 Interactive effects of sand burial and drought on shoot elongation of *B. argenteum***

Although *B. argenteum* shoots were generally less elongated in spring than that under the same treatment in autumn, drought and sand burial had negative and dual effects on shoot elongation, respectively, which was similar to the pattern observed for the other three parameters described above (Fig. 5). Conversely, drought reduced the positive effects of shallow burial and exacerbated the negative effects of deep burial (Fig. 5c,d) on shoot upgrowth, respectively. In addition, sand burial aggravated the negative effects of drought on shoot



elongation (Fig. 5a, b).

### 3.6 Redundancy analysis of the combined effects of sand burial and drought on the chlorophyll a content, PSII photochemical efficiency, regeneration potential, and shoot upgrowth of *B. argenteum*

The RDA analysis results showed that drought was a more important stressor influencing shoot elongation

than sand burial (Fig. 6), while sand burial played a more important role in the retention of the chlorophyll a content, PSII photochemical efficiency, and regeneration potential than drought. Specifically, sand burial was positively correlated with the chlorophyll a content, PSII photochemical efficiency, regeneration potential, and shoot elongation when the burial depth was shallow (< 2 mm, Fig. 6a), while it was negatively correlated with the four variables when the depth was larger (sand burial depth ≥ 2 mm, Fig. 6b). In addition, drought was

negatively correlated with the four variables when the burial depth was shallow (< 2 mm, Fig. 6a), but positively correlated with all variables, except for shoot elongation, when the depth was larger (sand burial depth ≥ 2 mm, Fig. 6b).

The four parameters investigated in this study were more readily affected by sand burial and drought in autumn than in spring when the burial depth was shallow (< 2 mm, Fig. 6a). Under deep sand burial (sand

burial depth ≥ 2 mm, Fig. 6b), the chlorophyll a content, PSII photochemical efficiency, and regeneration potential were more sensitive to sand burial in autumn than in spring, but shoot elongation was more susceptible to sand burial in spring than autumn. In contrast, the chlorophyll a content, PSII photochemical efficiency, and regeneration potential were more sensitive to drought in spring than in autumn, while shoot elongation was more susceptible to sand burial in autumn than spring under the deep sand burial treatments

(sand burial depth ≥ 2 mm, Fig. 6b).

### 4 Discussion

A desert is a multi-stressed environment, generally characterized by a series of stressors (Xie et al., 2007; Powell et al., 2015). The biocrust moss, *B. argenteum* Hedw. usually acts as a pioneer, and even dominant

species, inhabiting many desert ecosystems due to its high resistance and versatile adaptation strategies to

stressors (Li et al., 2014; Weber et al., 2016). There is growing evidence that biocrust organisms, including

mosses, are extremely vulnerable to stressors originating mostly from climate change and disturbances (Reed et

al., 2012; Weber et al., 2016). Ferrenberg et al. (2015) found that climate change and physical disturbances may

cause similar community shifts within biocrusts. In arid sandy desert ecosystems, drought and sand burial are

the two prevailing stressors, and are induced separately by climate change and disturbance. They act as filters,

eliminating the sensitive species by determining the physiology, growth, and survival of biocrust mosses

(Mart ñez and Maun, 1999; Barker et al., 2005; Jia et al., 2008). In this study, we found that drought and sand

burial exerted different, but dual effects on the physiology and growth of *B. argenteum*. More interestingly,

both antagonistic and additive effects of drought and sand burial on *B. argenteum* were observed (Fig. 6), which

explained the pattern of distribution of *B. argenteum* in an arid sandy desert where the two stressors can occur

simultaneously.

**4.1 Mutually antagonistic effects between drought and sand burial enabled *B. argenteum* to survive the co-occurrence of the two stressors in an arid sandy desert**

As hypothesized, it was found that a combination of sand burial and drought did not always exacerbate the

individual negative effects of each stressor on *B. argenteum*. Drought significantly ameliorated the negative

effects of deep sand burial on PSII photochemical efficiency (Fig. 2), the retention of chlorophyll a content (Fig.

3), and regeneration potential (Fig. 4) of *B. argenteum*. Sand burial diminished and even reversed the negative

effects of drought on the maintenance of the chlorophyll a content (Fig. 2), PSII photochemical efficiency (Fig.

3), and regeneration potential (Fig. 4) of *B. argenteum*. These mutually antagonistic effects on the physiological

vigor of the biocrust moss provided an opportunity for it to overcome the two co-occurring stressors, and this

may be an important reason why it usually acts as the pioneer moss species by colonizing and even flourishing

on the ground surface throughout China's sandy deserts.

The antagonistic effects of these two stressors are short-term physiological indicators, implying that *B.*





*argenteum* has a strong potential to photosynthesize or regenerate after their removal. It is difficult for the moss to maintain this potential for a long time due to the increased use or exhaustion of its stored carbohydrate reserves when buried (Maun, 1998; Kent et al., 2005). Therefore, other long-term parameters (e.g., growth rate) also need to be considered.

**4.2 Additive negative effects between drought and sand burial limit the distribution of *B. argenteum* in an arid sandy desert**

In contrast to our expectations, the mutually antagonistic effects of drought and sand burial did not impact on long-term shoot upgrowth, even though sand burial (depth ≤4 mm) stimulated shoot elongation (Fig. 5). This additive negative effect inflicted by the combination of drought and sand burial on shoot upgrowth suggested a

trade-off between growth and stress tolerance (Steinberg, 2012). In general, there is a tradeoff between growth and physiological vigor, including regeneration potential, when the moss is exposed to stress. This is in accordance with the theory that adaptation to stress carries a cost, and spending resources on defense or resistance leads to a weakened performance in conditions where these traits are not needed (Bijlsma and Loeschcke, 2005). Collectively, the preservation of physiological activity (photosynthetic pigment, PSII photochemical efficiency)

and the propagation of fecundity (protonemal area), afforded by the mutually antagonistic effects, at the cost of shoot elongation caused by the negative additive effect, under long-term, deep sand burial, will result in the failure of *B. argenteum* shoots to protrude above the sand surface. This could even lead to death, ultimately causing the moss to vanish from the ecosystem. This can explain the absence of *B. argenteum* in areas suffering from long-term, deep-sand-burial stress, such as flowing sand or seriously degraded landscapes. It also explains

why *B. argenteum* can colonize soil surfaces only after the burial depth decreases to a shallow level, through ecological construction and restoration measures throughout the arid sandy areas of northern China.

Based on a conceptual model, Bowker et al. (2006) and Li et al. (2010) both proved that biocrust moss is restricted to a specific topography, where it is less stressed from the microclimate and disturbances than at other positions at micro-spatial scales. However, the distribution of *B. argenteum* is apparently vaster and more





continuous than that indicated by Bowker et al. (2006) and Li et al. (2010) in arid sandy areas, where sand burial is pervasive and occurs regularly. Thus, it is suggested that the interaction between physical environmental stressors from resource limitation, climate, and physical disturbances can be used to facilitate an extension of the ecological niche of desert moss (Callaway, 1995). Therefore, corresponding models should take into account the interactions between climate change and physical disturbances, and from an evolutionary

perspective the environmental pressure and biological response should be considered integratively. This study also found that biocrust moss could be harmed by climate change, with conditions such as drought predicted to be more frequent and extensive in the future. This damage may be alleviated by other environmental factors or disturbances, such as sand burial, although this needs to be verified further.

**4.3 Possible mechanisms underlying the combined effects of drought and sand burial on *B. argenteum* and its significance in ecological construction**

It is not fully understood why drought and sand burial exert antagonistic effects on the physiological activity and asexual propagating fecundity of *B. argenteum*. It is possible that the antagonistic effect may originate from the water conserving effect of sand burial, which could mitigate the negative effect of drought to some extent

(Meng et al., 2011). For over 300 years in China, sand burial has been widely used by farmers as a useful moisture-conserving measure to cultivate crops, in a practice referred to as Shatian or sandy field (Li et al., 2000). However, this beneficial effect has rarely been reported for biocrust mosses. In addition, sand burial can also provide a protective shell for *B. argenteum*, mitigating the damage from other stresses, such as wind (Liu et al., 2013). On the other hand, drought favors *B. argenteum* under sand burial by lowering the risk of carbon

starvation induced by the reduction of photosynthetic area and the relatively trivial rainfall (causing the partial hydration of moss). This favorable effect of drought has also been reported for bundled filamentous cyanobacteria following sand burial (Williams, 2011; Rao et al., 2012) or thermal stress (Lan et al., 2014), and mosses exposed to heat shock (Xu et al., 2009), ultraviolet-B (Turnbull et al., 2009), and fungal attack (Weber et al., 2016). Drought would increase the removal rate of sand by enabling the dry sand to be more easily blown



by wind, resulting in a harmful deep burial becoming a beneficial shallow burial. Therefore, drought is also

considered to have a dual effect, especially when deep sand burial occurs.

       The ability to achieve a higher rate of shoot elongation gives *B. argenteum* an important advantage over

other moss species, enabling it to rapidly recover from sand burial (Jia et al., 2008). However, shoot elongation

in *B. argenteum* is severely inhibited by drought, due to the lower amounts (Jia et al., 2008) and/or shorter

durations (Kidron et al., 2010) of moisture availability. This could be interpreted as a reduction in the

accumulation of carbohydrate gained by photosynthesis or as carbon starvation (Barker et al., 2005) caused by

drought. Sand burial not only directly reduces the photosynthetic area of mosses, but also causes a deterioration

in the environmental conditions required for photosynthesis (e.g., reduction of photosynthetically active

radiation and blocking gas exchange). Furthermore, the sand deposited on *B. argenteum* would intercept water

from precipitation, decreasing the quantity of rainfall available to the moss, with the trivial amount of

precipitation received already identified as being detrimental (induce carbon starvation) to biocrust moss

(Alpert and Oechel, 1985; Belnap et al., 2004). This would consolidate the negative effect of drought on the

shoot elongation of *B. argenteum*.

       In recent years, the rapid artificial cultivation of biocrust has provided a novel alternative to traditional

biological methods for controlling erosion (Bu et al., 2014; Doherty et al., 2015; Antoninka et al., 2016). At the

same time, biocrust moss is considered to be a potentially promising biological material that could be

inoculated to accelerate the process of sand fixation and the recovery of degraded soil (Antoninka et al., 2016).

However, this technique is still limited to laboratory trials, with no successful large-scale application in the

field reported. One key reason for this is that the moss typically occupies the late successional stage among

biocrusts and its environmental requirements are high. The moss cultured in the laboratory under favorable

conditions cannot withstand the unfavorable stress from drought, high temperature, and UV-B exposure.

However, the results of this study indicate that moderate sand burial may have the potential to alleviate these

stresses and increase the survival ratio of artificially cultured biocrust moss in the restoration of arid sandy

deserts or degraded ecosystems. The results of our latest pilot experiment (unpublished) support this



proposition. The use of such a technique is also in agreement with Maestre et al. (2006), who found that the inoculation of biocrusts in the form of a slurry combined with the addition of composted sewage sludge, which has a similar effect to that of burial, encouraged the recovery of biocrust in degraded soils from semiarid Mediterranean areas.

**4.4 Seasonal effects on the combined effects of drought and sand burial on *B. argenteum***

Coverage (Table 1), physiological vigor (Fig. 2, 3), and growth rate (Fig. 4) of *B. argenteum* and the response to drought and sand burial varied with season (Fig. 5). In our study area, the physiological activity of *B. argenteum* reached its lowest level after a long-term cold and drought stress in winter (Li et al., 2012). Conserving its activity in the continuous-drought of spring, enabled the moss to be ready to obtain more carbon through photosynthesis in the relatively favorable conditions in summer (higher precipitation). In autumn, when the physiological activity of *B. argenteum* was highest, it is essential to gain height through shoot upgrowth to cope with the following sand burial in winter. This successful seasonal adaptation strategy of *B. argenteum* to the co-occurring stressors of drought and sand burial was supported by our results. Therefore, the seasonal distribution of precipitation and sand burial in our study area was important because it enabled *B. argenteum* to be the pioneer species in our study area, and this mechanism may be valid in areas suffering from the co-occurring stressors of drought and sand burial in sandy deserts elsewhere in China and worldwide.

*Acknowledgments*. This research was partially supported by the Natural Science Foundation of China (Grant Nos. 41371099, 41671210, 41621001) and CAS "Light of West China" Program.

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





**Table 1.** Changes in the percentage cover of a biocrust dominated by *Bryum. argenteum* in response to sand burial depth in spring and autumn.

| Sand burial | Season | |
| --- | --- | --- |
| depth (mm) | Spring | Autumn |
| 0 | 100.000±0.000a | 100.000±0.000a |
| 0.5 | 53.556±0.882c | 67.960±0.923b |
| 1 | 22.667±2.915e | 28.433±0.308d |
| 2 | 0.000±0.000g | 4.667±0.577f |
| 4 | 0.000±0.000g | 0.000±0.000g |
| 10 | 0.000±0.000g | 0.000±0.000g |

Values are means (± SE), different letters indicate significant differences between different sand burial depth treatments at the $p < 0.05$ level as determined using a least significant difference (LSD) post-hoc test, n = 9.




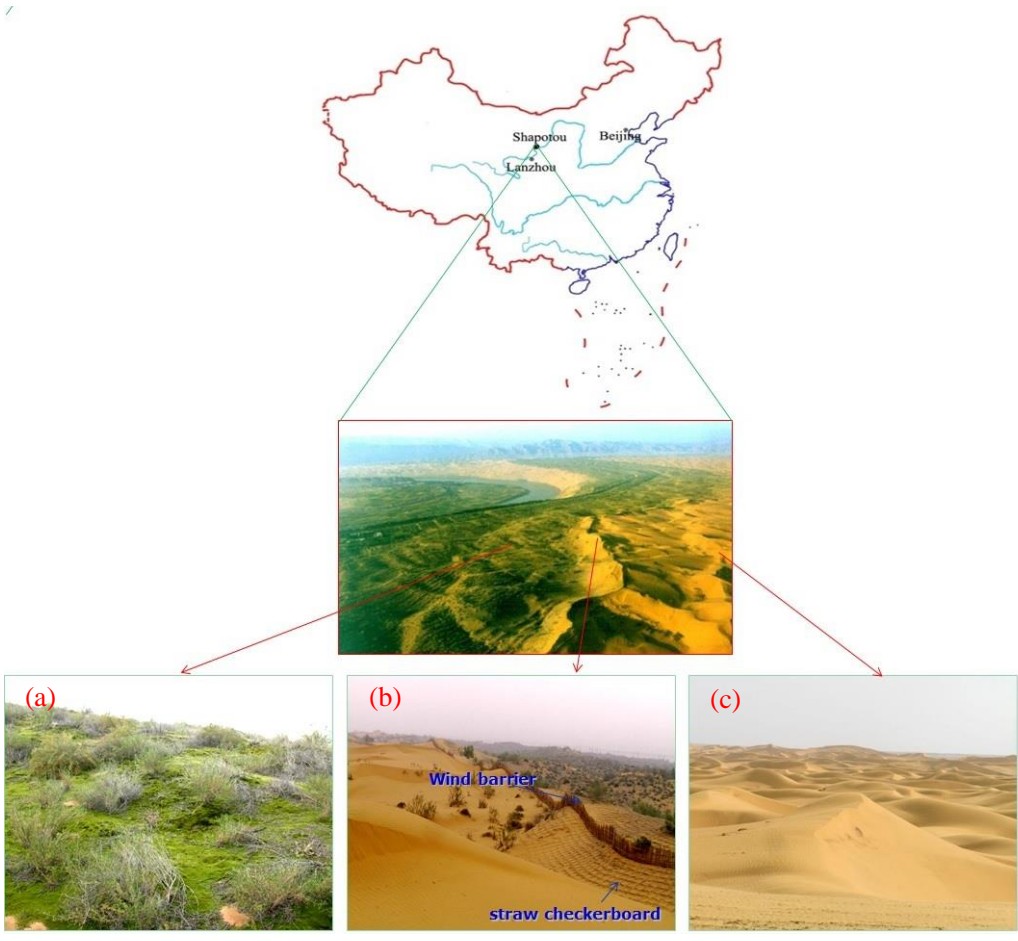

Figure 1. Location and main landscapes of the Shapotou zone in the southeastern edge of the Tengger Desert.

As a pioneer species, *Bryum argenteum* Hedw. has colonized and flourishes on the soil surface of the revegetation area (a) by controlling burial stress in the previously shifting sand dunes (c). This was initially achieved through the combined application of wind barriers, straw checkerboards, and the planting of shrubs without irrigation (b) 60 years ago.






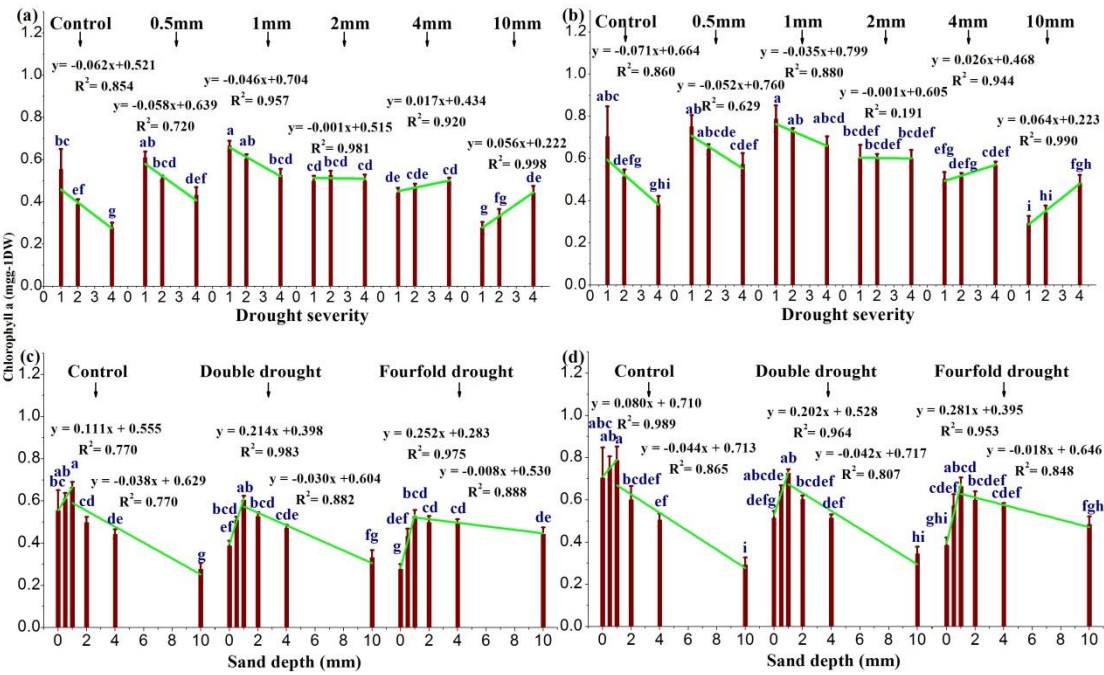

**Figure 2.** Changes in the chlorophyll a content of the biocrust moss *Bryum argenteum* Hedw. following exposure to a combination of three levels of drought severity and six depths of sand burial in spring (a, c) and autumn (b, d). Bars represent means ($\pm$ SE). Different letters indicate significant differences between different drought severities and sand burial depth treatments at the $p <0.05$ level, as determined using a least significant difference (LSD) post-hoc test.







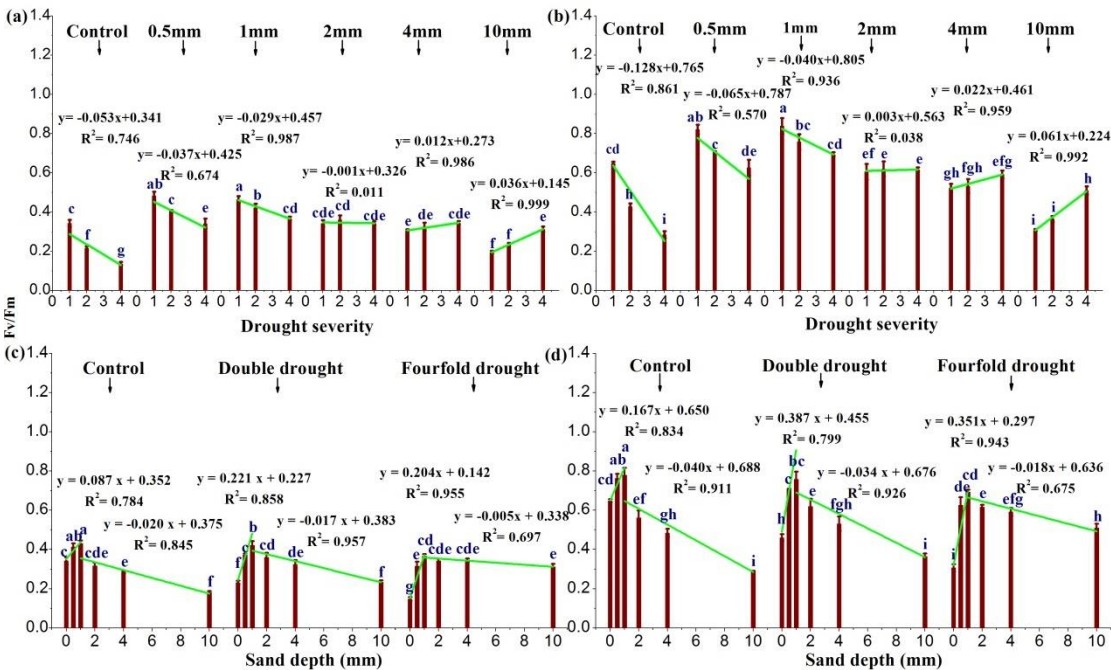


**Figure 3.** Changes in the PSII photochemical efficiency (Fv/Fm) of the biocrust moss *Bryum argenteum* Hedw. following exposure to a combination of three levels of drought severity and six depths of sand burial in spring (a, c) and autumn (b, d). Bars represent means ($\pm$ SE). Different letters indicate significant differences between different drought severities and sand burial depth treatments at the $p < 0.05$ level as determined using a least

significant difference (LSD) post-hoc test in spring and autumn.







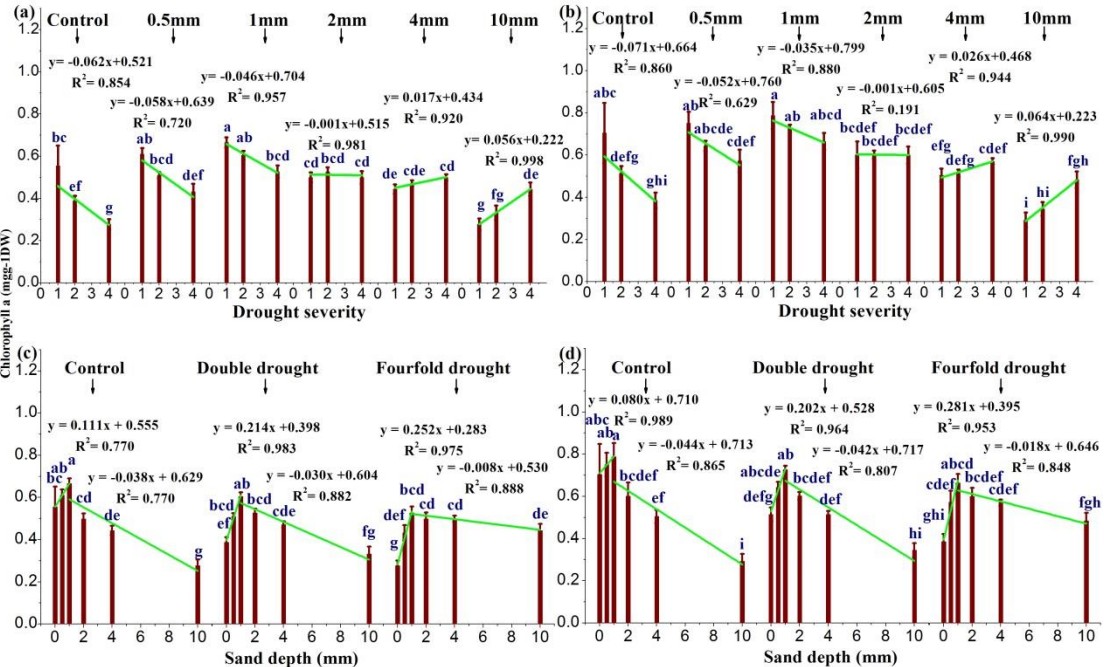

**Figure 4.** Changes in the protonemal area of detached shoots of biocrust moss *Bryum argenteum* Hedw.



following exposure to a combination of three levels of drought severity and six depths of sand burial in spring

(a, c) and autumn (b, d). Bars represent means (± SE). Different letters indicate significant differences between

different drought severities and sand burial depth treatments at the $p < 0.05$ level as determined using a least

significant difference (LSD) post-hoc test.







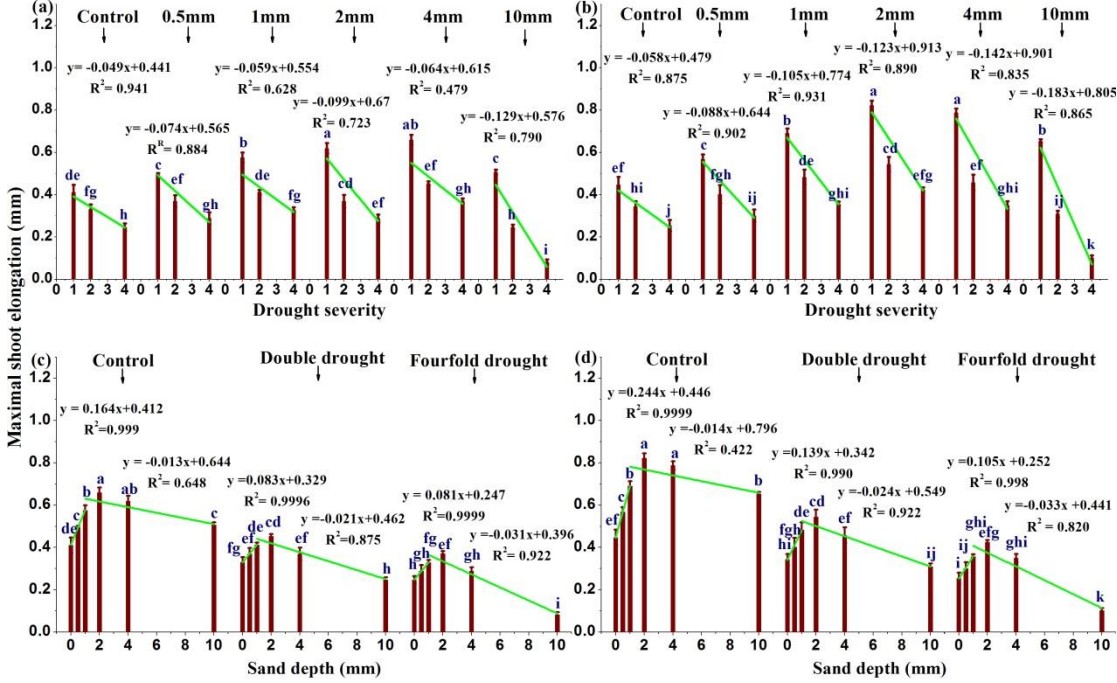

**Figure 5.** Changes in the maximal shoot elongation of biocrust moss *Bryum argenteum* Hedw. following exposure to a combination of three-severity drought and six-depth sand burial in spring (a, c) and autumn (b, d). Bars represent means ($\pm$ SE). Different letters indicate significant differences between different drought severities and sand burial depth treatments at the $p < 0.05$ level as determined using a least significant difference (LSD) post-hoc test.





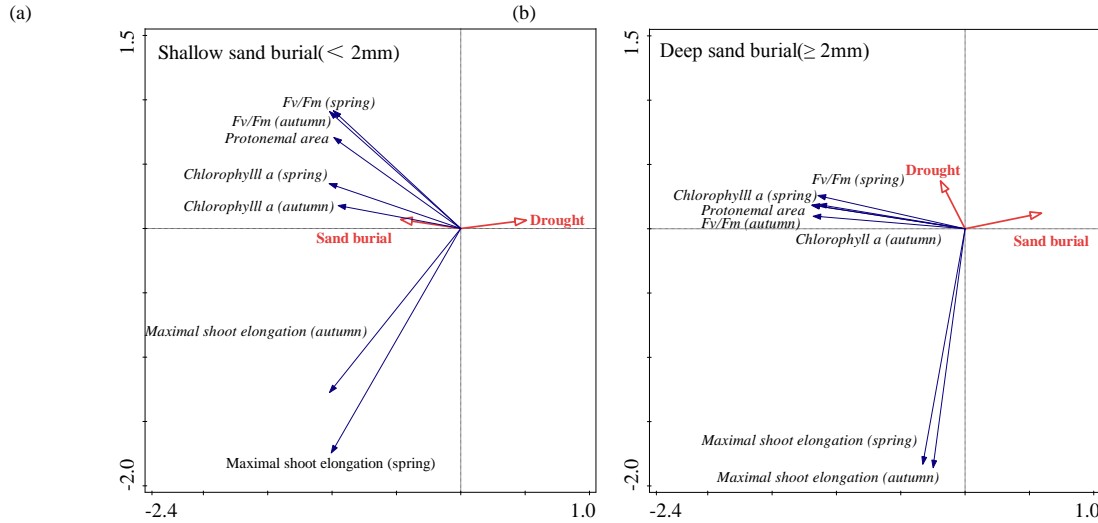

**Figure 6.** Redundancy analysis (RDA) diagram of the effect of drought, sand burial, and their combination on the chlorophyll a content, PSII photochemical efficiency (Fv/Fm), regeneration potential (protonemal area), and shoot upgrowth (maximal shoot elongation) of biocrust moss *Bryum argenteum* Hedw.

Under a shallow sand burial treatment, the eigenvalues were 0.8134 and 0.0152 for the first and second axes, respectively, and the correlation coefficients were 0.9527 and 0.4876, respectively. In terms of deep sand burial, the eigenvalues of the first and second axes were 0.6068 and 0.2379, respectively, and the correlation coefficients were 0.9362 and 0.9126, respectively. The Monte Carlo permutation test indicates that all variables were significantly correlated with the environmental factors ($p$ <0.05).