# Peer review of "Antagonistic effects of drought and sand burial enable survival of biocrust moss *Bryum* argenteum in an arid sandy desert"

_Biogeosciences, 2017_

## Referee Comment (RC1) · Anonymous Referee #1 · 10 Dec 2017

Journal: BG Title: The mutually antagonistic effect of drought and sand burial enables the biocrust moss Bryum argenteum Hedw. to survive the two co-occurring stressors in an arid sandy desert Author(s): Rongliang Jia et al. MS No.: bg-2017-402 MS Type: Research article Special Issue: Biological soil crusts and their role in biogeochemical processes and cycling

I found the manuscript by Rongliang Jia et al., interesting and fall within the scope of Biogeoscience Journal - under the sub-title "Plant– soil interactions", although its results, related to the combined effects of drought and sand burial on moss crust, are

particular interest to areas such as southeastern fringe of the Tengger Desert.

I have only minor comments:

I found their results sufficient to support the interpretations and conclusions. The title is too long and should be shortened.

Where are the Key words?

Moss not always represents the highest successional stage of the biocrust development. In most cases lichens as a slow growers are representing this stage, especially under dry conditions. However, climate with high rainfall may encourage moss growth in some areas such as the located in the southeastern fringe of the Tengger Desert.

Line 214 - the chlorophyll a content of argenteum was found should be B. argenteum.

Table 1. "Changes in the percentage cover of a biocrust dominated by Bryum. argenteum in response to sand burial depth in spring and autumn" should be " Table 1. Changes in the percentage cover of the soil surface, a biocrust dominated by Bryum argenteum in response to sand burial depth in spring and autumn".

Please also note the supplement to this comment:
https://www.biogeosciences-discuss.net/bg-2017-402/bg-2017-402-RC1-supplement.pdf

**Supplement:**

**The mutually antagonistic effect of drought and sand burial enables the biocrust moss *Bryum argenteum* Hedw. to survive the two co-occurring stressors in an arid sandy desert**
The title is too long and should be shorten.

Key words?

Moss not always represents the highest successional stage of the biocrust development. In most cases lichens as a slow growers are representing this stage, especially under dry conditions. However, climate with high rainfall may encourage moss growth in some areas such as the located in the southeastern fringe of the Tengger Desert.

214 the chlorophyll a content of argenteum was found should be B. argenteum.

Table 1. "Changes in the percentage cover of a biocrust dominated by Bryum. argenteum in response to sand burial depth in spring and autumn" should be " Table 1. Changes in the percentage cover of the soil surface, a biocrust dominated by Bryum argenteum in response to sand burial depth in spring and autumn".

I found the manuscript by Rongliang Jia et al., interesting and fall within the scope of Biogeoscience Journal, under the sub-title "Plant– soil interactions", although its results, related to the combined effects of drought and sand burial on moss crust, are particular interest to sandy desert areas such as southeastern fringe of the Tengger Desert. Sand burial in areas with movable sand dunes or drought were investigated separately, however their combine effects may suggest a novel approach.

I found their results sufficient to support the interpretations and conclusions.

---

## Referee Comment (RC2) · D. Elliott (Referee) · 2 Jan 2018

This paper presents research on the ability of biocrust mosses - in particular Bryum argenteum - to survive multiple stresses in dryland ecosystems. The concept is good and this research may ultimately support better land management and interventions, enabled by knowing the environmental controls on dryland biota. The novelty of the work is related to the simultaneous assessment of two stressors and their interactions: drought and burial. The experimental work appears to have been planned and carried out carefully with attention to detail, which gives me confidence in the results. The

results are presented in quite a confusing way though, which made it difficult for this reader to draw own interpretation and conclusions. A few details of the method were also not clear enough for me to fully understand the results, for example I was not sure whether the moss is completely buried or whether it sticks through some of the burial treatments, which is very important. A similar problem of detail and clarity affected reading of the discussion but to a lesser extent, and the final parts of the discussion were much clearer.

My suggestion for this paper is that the language and content should be revised with the aim of achieving clarity and detail relating to the objectives set out, and this may involve changing the figures too. Focusing discussion more on the fitness and adaptations of the moss is likely to help, and replacing commonly used vague terminology like "positive" and "negative" effects with specific observations or interpretations like "reduced chlorophyll content" or "increased fitness" will help further. I think that the experimental work is well conceived and of good quality, but at present it is hard to be sure whether the conclusions are fully supported by the results.

Specific comments below should help the authors see examples in the manuscript relating to the above suggestions:

8 "highest" is suggestive of being superior. Perhaps "latest" would be better?

11 surely this is a very large niche, not small as stated

45 Again use of "highest" does not seem appropriate

58 is there a reference to support this?

63 is there a reference to support this (that B. argenteum is usually the pioneer)?

70 "buried" perhaps a better word than "submerged"

72-76 here setting out the importance of understanding how B. argenteum survives these multiple stressors as a main theme in the work - an interesting objective with

practical applications.

85-90 it is stated that drought "protects" (benefit) and burial can "slow water loss" (benefit), so why are these then described as "mutually antagonistic" on line 87? In this context they are mutually beneficial. I think care is needed here (and earlier e.g. line 33) to note the difference between an apparently harsh environment (for humans) which is actually the niche for which certain biocrust organisms are adapted. Therefore, these harsh conditions are likely a requirement for life of the biocrust organisms being studied. Based on this one may reasonably assume that drought and burial are mutually beneficial for organisms adapted to live in this environment.

94-99 unnecessary precision of environmental parameters. The time period for which these data relate should be given.

126 the year should be given (and on line 133)

129 "below the ground surface" - where? In situ at the extraction place in the field, or somewhere else?

138 please explain the burial a bit more, and refer to this in the introduction and discussion too as appropriate. It is necessary to know whether the burial completely covers the moss, or whether it sticks up through the added sand. This has major implications for the interpretation and understanding of the work.

143 The experiment duration seems rather short but nothing can be done about that now. Perhaps the duration can be explained / justified?

147 Would deposited sand be naturally blown off in the field? The answer to this is of interest in relation to how the moss adapts to burial. If the deposit is never blown off then the moss needs to abandon the buried chlorophyll and invest in the tip, however if it might be blown off then it would make sense to retain the buried chlorophyll for a while in-case it will be exposed later.

152 A nice idea to minimise the edge effect.

159 Not clear if "shoot elongation" is the same as "shoot upgrowth" on line 157 - if same then please use same terminology, if different then please explain.

166 Again nice attention to detail in the experimental method here (randomising positions)

185 I think the unit is wrong here

211 and throughout results section: It is not clear what negative and positive effects are. Instead, just state which measurements changed and how they changed (e.g. decrease or increase in chlorphyll content). In general I found the results section quite difficult to follow because of this. Furthermore in some cases the language used unclear phrases (e.g. "decrease in autumn being significantly lower" line 206). There is some interpretation in results which should instead be in discussion.

284-286 I'm not sure if the effects observed have explained the moss distribution as claimed. Perhaps more explaination of this needed, or remove.

288 The discussion here is interesting, effectively summarising findings for the most part but still a bit confusing in relation to what is a positive or negative, which seems to partly contradict the introductory section e.g. lines 85-90

370-374 this is a clear and useful outcome of the work. Probably best not to mention the un-published results though.

380 This section also presenting clear and useful findings

Table 1. The precision to 3dp seems excessive, making it less clear

Figures 2-5. These are quite complicated and can't be fully understood based on the legend. For instance what is Control, 0.5mm, 1mm etc written at the top? What are the units or scale of drought severity? These details may be elsewhere but it should be possible to interpret the figures alone. I think some work is needed to make these a bit clearer, or if not possible consider using them as supplementary figures and replace

with something less detailed which is easier to interpret.

---

## Author Comment (AC2) · 3 Jan 2018

Referee's comment: This paper presents research on the ability of biocrust mosses - in particular Bryum argenteum - to survive multiple stresses in dryland ecosystems. The concept is good and this research may ultimately support better land management and interventions, enabled by knowing the environmental controls on dryland biota. The novelty of the work is related to the simultaneous assessment of two stressors and their interactions: drought and burial. The experimental work appears to have been planned and carried out carefully with attention to detail, which gives me confidence in

the results. The results are presented in quite a confusing way though, which made it difficult for this reader to draw own interpretation and conclusions. A few details of the method were also not clear enough for me to fully understand the results, for example I was not sure whether the moss is completely buried or whether it sticks through some of the burial treatments, which is very important. A similar problem of detail and clarity affected reading of the discussion but to a lesser extent, and the final parts of the discussion were much clearer. My suggestion for this paper is that the language and content should be revised with the aim of achieving clarity and detail relating to the objectives set out, and this may involve changing the figures too. Focusing discussion more on the fitness and adaptations of the moss is likely to help, and replacing commonly used vague terminology like "positive" and "negative" effects with specific observations or interpretations like "reduced chlorophyll content" or "increased fitness" will help further. I think that the experimental work is well conceived and of good quality, but at present it is hard to be sure whether the conclusions are fully supported by the results. Specific comments below should help the authors see examples in the manuscript relating to the above suggestions:

Authors' response: The language of our manuscript was improved by a European company (International Science Editing Compuscript Ltd.). The expressions were revised point by point according to the reviewer' suggestions to make them be more readable and understandable. In addition, we have carefully proof-read the manuscript to minimize typographical, grammatical, and bibliographical errors. Especially, we have revised the heading of Table 1 and Legends of Figure 2-5 as suggested, and from the data shown in Table 1, one can easily imagine that the mosses within the biocrust were only partially buried by sand when the depth of burial treated was shallow (depth $\leq$2 mm), while, the mosses were completely buried when the burial depth$\geq$ 4mm. We agree that the specific observations or interpretations like "reduced chlorophyll content" or "increased fitness" will help increase the clarity of the results, we used the similar ones although very few. Considering the major objective/focus of this manuscript is to explore the interaction between drought and sand burial on biocrust moss, ie., to make

sure whether their interaction is additive or antagonistic, so the terminology "positive" and "negative" effects were used. While, we have made many modifications in Results to improve its understandability.

Referee's comment: 8 "highest" is suggestive of being superior. Perhaps "latest" would be better? 45 Again use of "highest" does not seem appropriate

Authors' response: Yes, we replaced "highest" with "latest" in L8 and L46.

Referee's comment: 11 surely this is a very large niche, not small as stated

Authors' response: We replaced "small" with "very large" in L11.

Referee's comment: 58 is there a reference to support this?

Authors' response: We have added a reference, "Li et al., 2004", to support this.

Referee's comment: 63 is there a reference to support this (that B. argenteum is usually the pioneer)?

Authors' response: This finding derives from our one recent field investigation conducted throughout China's desert, which will be exhibited in our next paper. As far as I know, there is no published reference in English to support this to date.

Referee's comment: 70 "buried" perhaps a better word than "submerged"

Authors' response: We replaced "submerged" with "buried" in L72.

Referee's comment: 72-76 here setting out the importance of understanding how B. argenteum survives these multiple stressors as a main theme in the work - an interesting objective with practical applications.

Authors' response: I agree, while we are convinced that the original sentences is enough to express the meaning, so we did not add new sentences.

Referee's comment: 85-90 it is stated that drought "protects" (benefit) and burial can "slow water loss" (benefit), so why are these then described as "mutually antagonistic"

on line 87? In this context they are mutually beneficial. I think care is needed here (and earlier e.g. line 33) to note the difference between an apparently harsh environment (for humans) which is actually the niche for which certain biocrust organisms are adapted. Therefore, these harsh conditions are likely a requirement for life of the biocrust organisms being studied. Based on this one may reasonably assume that drought and burial are mutually beneficial for organisms adapted to live in this environment.

Authors' response: Drought and sand burial are two commonly-viewed stressors that severely limit (even damage in most cases by intuition) the growth and distribution of biocrust moss. Here we want to show that they can also induce beneficial effects in some cases, respectively, which would potentially introduce beneficial effects by the combination of some specific level of drought and a certain depth of sand burial.

Referee's comment: 94-99 unnecessary precision of environmental parameters. The time period for which these data relate should be given.

Authors' response: We added "Based on meteorological records from 1956 to 2003" in L98.

Referee's comment: 126 the year should be given (and on line 133) Authors' response: We added "in 2013" in L131.

Referee's comment: 129 "below the ground surface" - where? In situ at the extraction place in the field, or somewhere else?

Authors' response: Yes, we added "and transferred to Water Balance Observation Site (about 1 km away from the sampling place) in Shapotou Desert Research and Experiment Station, Chinese Academy of Sciences." in L128-129.

Referee's comment: 138 please explain the burial a bit more, and refer to this in the introduction and discussion too as appropriate. It is necessary to know whether the burial completely covers the moss, or whether it sticks up through the added sand. This has major implications for the interpretation and understanding of the work.

Authors' response: I think the burial phenomena can be easily and clearly imagined from the data shown in Table 1(similar to Jia et al., Soil Biol. Biochem., 40, 2827-2834, 2008; Martínez, Plant Ecol., 145, 209-219, 1999; Maun, M. A. Can. J. Bot., 7, 713–738, 1998; Maun, M. A. Coastal Dunes: Ecology and Management, Springer, 119–136, 2008.), that the mosses within the biocrust were only partially buried by sand when the depth of burial treated was shallow (depth $\leq$2 mm), while, the mosses were completely buried when the burial depth$\geq$ 4mm. To avoid misunderstanding, we have revised the heading of Table 1 to "Table 1. Changes in the percentage cover of Bryum argenteum Hedw. within a biocrust in response to sand burial depth in spring and autumn.".

Referee's comment: 143 The experiment duration seems rather short but nothing can be done about that now. Perhaps the duration can be explained / justified?

Authors' response: Yes, 72 d is short for each season. We did our best to make it close to real time of each season and to keep them the same in both spring and autumn, but we are convinced that it is enough for this study.

Referee's comment: 147 Would deposited sand be naturally blown off in the field? The answer to this is of interest in relation to how the moss adapts to burial. If the deposit is never blown off then the moss needs to abandon the buried chlorophyll and invest in the tip, however if it might be blown off then it would make sense to retain the buried chlorophyll for a while in-case it will be exposed later.

Authors' response: Yes, the sand deposited above the biocrust moss can be blown off in nature. We had discussed this in L357-358 in Discussion.

Referee's comment: 152 A nice idea to minimise the edge effect. 159 Not clear if "shoot elongation" is the same as "shoot upgrowth" on line 157 – if same then please use same terminology, if different then please explain.

Authors' response: For biocrust moss under sand burial, according to our observation, "shoot elongation" and "shoot upgrowth" have the same meaning.

Referee's comment: 166 Again nice attention to detail in the experimental method here (randomising positions) 185 I think the unit is wrong here Authors' response: We replaced "g" with "rpm" in L187-188.

Referee's comment: 211 and throughout results section: It is not clear what negative and positive effects are. Instead, just state which measurements changed and how they changed (e.g. decrease or increase in chlorphyll content). In general I found the results section quite difficult to follow because of this. Furthermore in some cases the language used unclear phrases (e.g. "decrease in autumn being significantly lower" line 206). There is some interpretation in results which should instead be in discussion.

Authors' response: To improve the understandability, we revised the original sentences to "Drought uniformly imposed negative effects (the slopes of the fitted lines were negative) on the chlorophyll a content (Fig. 2a, b), whereas burial by sand had a dual effect (the slopes of the fitted lines both have positive and negative values) on the chlorophyll a content (Fig. 2c, d)."in L215-217,"Drought consistently exerted negative effects (the slopes of the fitted lines were negative) on the PSII photochemical efficiency (Fig. 3a, b), while burial by sand had a dual effect (the slopes of the fitted lines both exhibited positive and negative values) on the PSII photochemical efficiency (Fig. 3c, d)."in L229-232, "Drought imposed negative effects (the slopes of the fitted lines were negative) on the regeneration potential of detached shoots of B. argenteum (Fig. 4a), while burial by sand had a dual effect (the slopes of the fitted lines both displayed positive and negative values) on the regeneration potential (Fig. 4b)." in L242-244 and "Although B. argenteum shoots were generally less elongated in spring than that under the same treatment in autumn, drought and sand burial, according to the slope values of the corresponding fitted lines, had negative and dual effects on shoot elongation, respectively"in L252-253. In addition, we replaced "decrease" with "amplitude decreased" in L209.

Referee's comment: 284-286 I'm not sure if the effects observed have explained the moss distribution as claimed. Perhaps more explaination of this needed, or remove.

Authors' response: To be more accurate, we replaced "pattern of" with "survival and" in L293.

Referee's comment: 288 The discussion here is interesting, effectively summarising findings for the most part but still a bit confusing in relation to what is a positive or negative, which seems to partly contradict the introductory section e.g. lines 85-90

Authors' response: We think that there is no contradiction and can be explained. The few research cases listed in Introduction showed that drought and sand burial can separately impose only one-way, beneficial effects on biocrust moss besides harmful influences by intuition. While, this study showed that, drought and sand burial singly had more complex, dual effects in most cases, the beneficial and detrimental effects caused by either factor can emerge simultaneously and shift along with the changes in the severity of drought or depth of burial. These indisputably introduced more complicated effects to biocrust moss when then combined. Even though, we are convinced that we have clearly displayed their single or interactive influences on every parameter tested of B. argenteum.

Referee's comment: 370-374 this is a clear and useful outcome of the work. Probably best not to mention the un-published results though.

Authors' response: We deleted the sentence with the un-published results.

Referee's comment: 380 This section also presenting clear and useful findings Table 1. The precision to 3dp seems excessive, making it less clear

Authors' response: We have redrawn Table 1 to make it be clearer.

Referee's comment: Figures 2-5. These are quite complicated and can't be fully understood based on the legend. For instance what is Control, 0.5mm, 1mm etc written at the top? What are the units or scale of drought severity? These details may be elsewhere but it should be possible to interpret the figures alone. I think some work is needed to make these a bit clearer, or if not possible consider using them as supplementary figures and replace with something less detailed which is easier to interpret.

Authors' response: To improve the clarity and understandability, we have revised the legends of Figures 2-5, as shown below:

Figure 2. Changes in the chlorophyll a content of the biocrust moss Bryum argenteum Hedw. following exposure to the combinations of single (receiving natural precipitation amount, control), double (receiving 1/2 of natural precipitation amount) and fourfold (receiving 1/4 of natural precipitation amount) drought and 0 (control), 0.5, 1, 2, 4 and 10 mm depths of sand burial in spring (a, c) and autumn (b, d). Bars represent means ($\pm$ SE). Different letters indicate significant differences between different drought severities and sand burial depth treatments at the p <0.05 level, as determined using a least significant difference (LSD) post-hoc test.

Figure 3. Changes in the PSII photochemical efficiency (Fv/Fm) of the biocrust moss Bryum argenteum Hedw. following exposure to the combinations of single (receiving natural precipitation amount, control), double (receiving 1/2 of natural precipitation amount) and fourfold (receiving 1/4 of natural precipitation amount) drought and 0 (control), 0.5, 1, 2, 4 and 10 mm depths of sand burial in spring (a, c) and autumn (b, d). Bars represent means ($\pm$ SE). Different letters indicate significant differences between different drought severities and sand burial depth treatments at the p <0.05 level as determined using a least significant difference (LSD) post-hoc test in spring and autumn.

Figure 4. Changes in the protonemal area of detached shoots of biocrust moss Bryum argenteum Hedw. following exposure to the combinations of single (receiving natural precipitation amount, control), double (receiving 1/2 of natural precipitation amount) and fourfold (receiving 1/4 of natural precipitation amount) drought and 0 (control), 0.5, 1, 2, 4 and 10 mm depths of sand burial in spring. Bars represent means ($\pm$ SE). Different letters indicate significant differences between different drought severities and sand burial depth treatments at the p <0.05 level as determined using a least significant difference (LSD) post-hoc test.

Figure 5. Changes in the maximal shoot elongation of biocrust moss Bryum argenteum Hedw. following exposure to the combinations of single (receiving natural precipitation amount, control), double (receiving 1/2 of natural precipitation amount) and fourfold (receiving 1/4 of natural precipitation amount) drought and 0 (control), 0.5, 1, 2, 4 and 10 mm depths of sand burial in spring (a, c) and autumn (b, d). Bars represent means ($\pm$ SE). Different letters indicate significant differences between different drought severities and sand burial depth treatments at the $p < 0.05$ level as determined using a least significant difference (LSD) post-hoc test.

In addition, Figure 4 was replaced with the right one.

All these responses and revised manuscript with the changes marked please see the supplement file.

Please also note the supplement to this comment:
https://www.biogeosciences-discuss.net/bg-2017-402/bg-2017-402-AC2-supplement.pdf

**Supplement:**

**Authors' response to comments of reviewer # 2**

**Referee's comment:**

This paper presents research on the ability of biocrust mosses - in particular *Bryum argenteum* - to survive multiple stresses in dryland ecosystems. The concept is good and this research may ultimately support better land management and interventions, enabled by knowing the environmental controls on dryland biota. The novelty of the work is related to the simultaneous assessment of two stressors and their interactions: drought and burial. The experimental work appears to have been planned and carried out carefully with attention to detail, which gives me confidence in the results. The results are presented in quite a confusing way though, which made it difficult for this reader to draw own interpretation and conclusions. A few details of the method were also not clear enough for me to fully understand the results, for example I was not sure whether the moss is completely buried or whether it sticks through some of the burial treatments, which is very important. A similar problem of detail and clarity affected reading of the discussion but to a lesser extent, and the final parts of the discussion were much clearer. My suggestion for this paper is that the language and content should be revised with the aim of achieving clarity and detail relating to the objectives set out, and this may involve changing the figures too. Focusing discussion more on the fitness and adaptations of the moss is likely to help, and replacing commonly used vague terminology like "positive" and "negative" effects with specific observations or interpretations like "reduced chlorophyll content" or "increased fitness" will help further. I think that the experimental work is well conceived and of good quality, but at present it is hard to be sure whether the conclusions are fully supported by the results. Specific comments below should help the authors see examples in the manuscript relating to the above suggestions:

*Authors' response:*

*The language of our manuscript was improved by a European company (International Science Editing Compuscript Ltd.). The expressions were revised point by point according to the reviewer' suggestions to make them be more readable and understandable. In addition, we have carefully proof-read the manuscript to minimize typographical, grammatical, and bibliographical errors. Especially, we have revised the heading of Table 1 and Legends of Figure 2-5 as suggested, and from the data shown in Table 1, one can easily imagine that the mosses within the biocrust were only partially buried by sand when the depth of burial treated was shallow (depth ≤2 mm), while, the mosses were completely buried when the burial depth≥ 4mm.*

*We agree that the specific observations or interpretations like "reduced chlorophyll content" or "increased fitness" will help increase the clarity of the results, we used the similar ones although very few. Considering the major objective/focus of this manuscript is to explore the interaction between drought and sand burial on biocrust moss, ie., to make sure whether their interaction is additive or antagonistic, so the terminology "positive" and "negative" effects were used. While, we have made many modifications in Results to improve its understandability.*

**Referee's comment:**

8 "highest" is suggestive of being superior. Perhaps "latest" would be better?

45 Again use of "highest" does not seem appropriate

*Authors' response*:

*Yes, we replaced "highest" with "latest" in L8 and L46.*

**Referee's comment:**

11 surely this is a very large niche, not small as stated

*Authors' response*:

*We replaced "small" with "very large" in L11.*

**Referee's comment:**

58 is there a reference to support this?

*Authors' response*:

*We have added a reference, "Li et al., 2004", to support this.*

**Referee's comment:**

63 is there a reference to support this (that B. argenteum is usually the pioneer)?

*Authors' response*:

*This finding derives from our one recent field investigation conducted throughout China's desert, which will be exhibited in our next paper. As far as I know, there is no published reference in English to support this to date.*

**Referee's comment:**

70 "buried" perhaps a better word than "submerged"

*Authors' response*:

*We replaced "submerged" with "buried" in L72.*

**Referee's comment:**

72-76 here setting out the importance of understanding how B. argenteum survives these multiple stressors as a main theme in the work - an interesting objective with practical applications.

*Authors' response*:

*I agree, while we are convinced that the original sentences is enough to express the meaning, so we did not add new sentences.*

**Referee's comment:**

85-90 it is stated that drought "protects" (benefit) and burial can "slow water loss" (benefit), so why are these then described as "mutually antagonistic" on line 87? In this context they are mutually beneficial. I think care is needed here (and earlier e.g. line 33) to note the difference between an apparently harsh environment (for

humans) which is actually the niche for which certain biocrust organisms are adapted. Therefore, these harsh conditions are likely a requirement for life of the biocrust organisms being studied. Based on this one may reasonably assume that drought and burial are mutually beneficial for organisms adapted to live in this environment.

*Authors' response***:**

*Drought and sand burial are two commonly-viewed stressors that severely limit (even damage in most cases by intuition) the growth and distribution of biocrust moss. Here we want to show that they can also induce beneficial effects in some cases, respectively, which would potentially introduce beneficial effects by the combination of some specific level of drought and a certain depth of sand burial.*

**Referee's comment:**

94-99 unnecessary precision of environmental parameters. The time period for which these data relate should be given.

*Authors' response***:**

*We added "Based on meteorological records from 1956 to 2003" in L98.*

**Referee's comment:**

126 the year should be given (and on line 133)

*Authors' response***:**

*We added "in 2013" in L131.*

**Referee's comment:**

129 "below the ground surface" - where? In situ at the extraction place in the field, or somewhere else?

*Authors' response***:**

*Yes, we added "and transferred to Water Balance Observation Site (about 1 km away from the sampling place) in Shapotou Desert Research and Experiment Station, Chinese Academy of Sciences." in L128-129.*

**Referee's comment:**

138 please explain the burial a bit more, and refer to this in the introduction and discussion too as appropriate. It is necessary to know whether the burial completely covers the moss, or whether it sticks up through the added sand. This has major implications for the interpretation and understanding of the work.

*Authors' response***:**

*I think the burial phenomena can be easily and clearly imagined from the data shown in Table 1(similar to Jia et al., Soil Biol. Biochem., 40, 2827-2834, 2008; Mart ńez, Plant Ecol., 145, 209-219, 1999; Maun, M. A. Can. J. Bot., 7, 713–738, 1998; Maun, M. A. Coastal Dunes: Ecology and Management, Springer, 119–136, 2008.), that the mosses within the biocrust were only partially buried by sand when the depth of burial treated was shallow (depth ≤2 mm), while, the mosses were completely buried when the burial depth≥ 4mm. To avoid*

*misunderstanding, we have revised the heading of Table 1 to "Table 1. Changes in the percentage cover of Bryum argenteum Hedw. within a biocrust in response to sand burial depth in spring and autumn.".*

**Referee's comment:**

143 The experiment duration seems rather short but nothing can be done about that now. Perhaps the duration can be explained / justified?

*Authors' response:*

*Yes, 72 d is short for each season. We did our best to make it close to real time of each season and to keep them the same in both spring and autumn, but we are convinced that it is enough for this study.*

**Referee's comment:**

147 Would deposited sand be naturally blown off in the field? The answer to this is of interest in relation to how the moss adapts to burial. If the deposit is never blown off then the moss needs to abandon the buried chlorophyll and invest in the tip, however if it might be blown off then it would make sense to retain the buried chlorophyll for a while in-case it will be exposed later.

*Authors' response:*

*Yes, the sand deposited above the biocrust moss can be blown off in nature. We had discussed this in L357-358 in Discussion.*

**Referee's comment:**

152 A nice idea to minimise the edge effect.

159 Not clear if "shoot elongation" is the same as "shoot upgrowth" on line 157 – if same then please use same terminology, if different then please explain.

*Authors' response:*

*For biocrust moss under sand burial, according to our observation, "shoot elongation" and "shoot upgrowth" have the same meaning.*

**Referee's comment:**

166 Again nice attention to detail in the experimental method here (randomising positions)

185 I think the unit is wrong here

*Authors' response:*

*We replaced "g" with "rpm" in L187-188.*

**Referee's comment:**

211 and throughout results section: It is not clear what negative and positive effects are. Instead, just state which measurements changed and how they changed (e.g. decrease or increase in chlorophyll content). In general I found the results section quite difficult to follow because of this. Furthermore in some cases the

language used unclear phrases (e.g. "decrease in autumn being significantly lower" line 206). There is some interpretation in results which should instead be in discussion.

*Authors' response*:

*To improve the understandability, we revised the original sentences to "Drought uniformly imposed negative effects (the slopes of the fitted lines were negative) on the chlorophyll a content (Fig. 2a, b), whereas burial by sand had a dual effect (the slopes of the fitted lines both have positive and negative values) on the chlorophyll a content (Fig. 2c, d)."in L215-217,"Drought consistently exerted negative effects (the slopes of the fitted lines were negative) on the PSII photochemical efficiency (Fig. 3a, b), while burial by sand had a dual effect (the slopes of the fitted lines both exhibited positive and negative values) on the PSII photochemical efficiency (Fig. 3c, d)."in L229-232, "Drought imposed negative effects (the slopes of the fitted lines were negative) on the regeneration potential of detached shoots of B. argenteum (Fig. 4a), while burial by sand had a dual effect (the slopes of the fitted lines both displayed positive and negative values) on the regeneration potential (Fig. 4b)." in L242-244 and "Although B. argenteum shoots were generally less elongated in spring than that under the same treatment in autumn, drought and sand burial, according to the slope values of the corresponding fitted lines, had negative and dual effects on shoot elongation, respectively"in L252-253.*

*In addition, we replaced "decrease" with "amplitude decreased" in L209.*

**Referee's comment:**

284-286 I'm not sure if the effects observed have explained the moss distribution as claimed. Perhaps more explaination of this needed, or remove.

*Authors' response*:

*To be more accurate, we replaced "pattern of" with "survival and" in L293.*

**Referee's comment:**

288 The discussion here is interesting, effectively summarising findings for the most part but still a bit confusing in relation to what is a positive or negative, which seems to partly contradict the introductory section e.g. lines 85-90

*Authors' response*:

*We think that there is no contradiction and can be explained. The few research cases listed in Introduction showed that drought and sand burial can separately impose only one-way, beneficial effects on biocrust moss besides harmful influences by intuition. While, this study showed that, drought and sand burial singly had more complex, dual effects in most cases, the beneficial and detrimental effects caused by either factor can emerge simultaneously and shift along with the changes in the severity of drought or depth of burial. These indisputably introduced more complicated effects to biocrust moss when then combined. Even though, we are convinced that we have clearly displayed their single or interactive influences on every parameter tested of B. argenteum.*

**Referee's comment:**

370-374 this is a clear and useful outcome of the work. Probably best not to mention the un-published results though.

***Authors' response*:**

*We deleted the sentence with the un-published results.*

**Referee's comment:**

380 This section also presenting clear and useful findings

Table 1. The precision to 3dp seems excessive, making it less clear

***Authors' response*:**

*We have redrawn Table 1 to make it be clearer.*

**Referee's comment:**

Figures 2-5. These are quite complicated and can't be fully understood based on the legend. For instance what is Control, 0.5mm, 1mm etc written at the top? What are the units or scale of drought severity? These details may be elsewhere but it should be possible to interpret the figures alone. I think some work is needed to make these a bit clearer, or if not possible consider using them as supplementary figures and replace with something less detailed which is easier to interpret.

***Authors' response*:**

*To improve the clarity and understandability, we have revised the legends of Figures 2-5, as shown below:*

***Figure 2.** Changes in the chlorophyll a content of the biocrust moss Bryum argenteum Hedw. following exposure to the combinations of single (receiving natural precipitation amount, control), double (receiving 1/2 of natural precipitation amount) and fourfold (receiving 1/4 of natural precipitation amount) drought and 0 (control), 0.5, 1, 2, 4 and 10 mm depths of sand burial in spring (a, c) and autumn (b, d). Bars represent means ($\pm$ SE). Different letters indicate significant differences between different drought severities and sand burial depth treatments at the $p <0.05$ level, as determined using a least significant difference (LSD) post-hoc test.*

***Figure 3.** Changes in the PSII photochemical efficiency (Fv/Fm) of the biocrust moss* Bryum argenteum *Hedw. following exposure to the combinations of single (receiving natural precipitation amount, control), double (receiving 1/2 of natural precipitation amount) and fourfold (receiving 1/4 of natural precipitation amount) drought and 0 (control), 0.5, 1, 2, 4 and 10 mm depths of sand burial in spring (a, c) and autumn (b, d). Bars represent means ($\pm$ SE). Different letters indicate significant differences between different drought severities and sand burial depth treatments at the* $p <0.05$ *level as determined using a least significant difference (LSD) post-hoc test in spring and autumn.*

***Figure 4.** Changes in the protonemal area of detached shoots of biocrust moss* Bryum argenteum *Hedw. following exposure to the combinations of single (receiving natural precipitation amount, control), double (receiving 1/2 of natural precipitation amount) and fourfold (receiving 1/4 of natural precipitation amount) drought and 0 (control), 0.5, 1, 2, 4 and 10 mm depths of sand burial in spring. Bars represent means ($\pm$ SE). Different letters indicate significant differences between different drought severities and sand burial depth*

treatments at the p <0.05 level as determined using a least significant difference (LSD) post-hoc test.

**Figure 5.** *Changes in the maximal shoot elongation of biocrust moss* Bryum argenteum *Hedw. following exposure to the combinations of single (receiving natural precipitation amount, control), double (receiving 1/2 of natural precipitation amount) and fourfold (receiving 1/4 of 
[revised manuscript text omitted]
 the combinations of single (receiving natural precipitation amount, control), double (receiving 1/2 of natural

precipitation amount) and fourfold (receiving 1/4 of natural precipitation amount) drought and 0 (control), 0.5,

555

1, 2, 4 and 10 mm depths of sand burial in spring (a, c)

and autumn (b, d). Bars represent means ($\pm$ SE). Different letters indicate significant differences between

different drought severities and sand burial depth treatments at the $p < 0.05$ level, as determined using a least

significant difference (LSD) post-hoc test.

560

[Figure]

**Figure 3.** Changes in the PSII photochemical efficiency (Fv/Fm) of the biocrust moss *Bryum argenteum* Hedw.

following exposure to the combinations of single (receiving natural precipitation amount, control), double

 (receiving 1/2 of natural precipitation amount) and fourfold (receiving 1/4 of natural precipitation amount)

drought and 0 (control), 0.5, 1, 2, 4 and 10 mm depths of sand burial in spring (a, c) and autumn (b, d). Bars represent means (± SE). Different letters indicate significant differences between different drought severities and sand burial depth treatments at the $p <0.05$ level as determined using a least significant difference (LSD) post-hoc test in spring and autumn.

580

585

590

[Figure]

[Figure]

**Figure 4.** Changes in the protonemal area of detached shoots of biocrust moss *Bryum argenteum* Hedw. following exposure to the combinations of single (receiving natural precipitation amount, control), double (receiving 1/2 of natural precipitation amount) and fourfold (receiving 1/4 of natural precipitation amount) drought and 0 (control), 0.5, 1, 2, 4 and 10 mm depths of sand burial in spring . Bars represent means (± SE). Different letters indicate significant differences between different drought severities and sand burial depth treatments at the *p* <0.05 level as determined using a least significant difference (LSD) post-hoc test.

[Figure]

**Figure 5.** Changes in the maximal shoot elongation of biocrust moss *Bryum argenteum* Hedw. following exposure to the combinations of single (receiving natural precipitation amount, control), double (receiving 1/2

630   of natural precipitation amount) and fourfold (receiving 1/4 of natural precipitation amount) drought and 0
(control), 0.5, 1, 2, 4 and 10 mm depths of  sand burial in
spring (a, c) and autumn (b, d). Bars represent means ($\pm$ SE). Different letters indicate significant differences
between different drought severities and sand burial depth treatments at the $p < 0.05$ level as determined using a
least significant difference (LSD) post-hoc test.

635

640

645

(a)                                                        (b)

[Figure]

**Figure 6.** Redundancy analysis (RDA) diagram of the effect of drought, sand burial, and their combination on the chlorophyll a content, PSII photochemical efficiency (Fv/Fm), regeneration potential (protonemal area), and shoot upgrowth (maximal shoot elongation) of biocrust moss *Bryum argenteum* Hedw.

Under a shallow sand burial treatment, the eigenvalues were 0.8134 and 0.0152 for the first and second axes, respectively, and the correlation coefficients were 0.9527 and 0.4876, respectively. In terms of deep sand burial, the eigenvalues of the first and second axes were 0.6068 and 0.2379, respectively, and the correlation coefficients were 0.9362 and 0.9126, respectively. The Monte Carlo permutation test indicates that all variables were significantly correlated with the environmental factors ($p < 0.05$).

---

## Author Response (AR1)

**Point-by-point response to the reviews**

**Part A: Authors' response to comments of reviewer # 1**

**Referee's comment:**

I found the manuscript by Rongliang Jia et al., interesting and fall within the scope of Biogeoscience Journal - under the sub-title "Plant– soil interactions", although its results, related to the combined effects of drought and sand burial on moss crust, are particular interest to areas such as southeastern fringe of the Tengger Desert.

I have only minor comments:

I found their results sufficient to support the interpretations and conclusions. The title is too long and should be shortened.

*Authors' response:*

*Because Bryum argenteum is widely distributed throughout the world, our findings could also be extended to ecosystem restoration and management in deserts beyond China.*

*The title was shortened to "The mutually antagonistic effect of drought and sand burial enables the biocrust moss Bryum argenteum Hedw. to survive in an arid sandy desert"*

**Referee's comment:**

Where are the Key words?

*Authors' response:*

*We have added "antagonistic effect, drought, sand burial, Bryum argenteum" as Key words in the new version of the manuscript.*

**Referee's comment:**

Moss not always represents the highest successional stage of the biocrust development. In most cases lichens as a slow growers are representing this stage, especially

under dry conditions. However, climate with high rainfall may encourage moss growth

in some areas such as the located in the southeastern fringe of the Tengger Desert.

*Authors' response:*

*It is true that a moss crust is not always the highest successional stage of biocrust development throughout arid and semiarid regions.*

*We revised the introductory sentences to, "It can represent the highest succession stage among the diverse range of surface-dwelling cryptogams (e.g., cyanobacteria, green algae, and lichen, which are also referred to as biocrusts) and it can make a major contribution to soil stability and fertility in many arid sandy desert ecosystems." in L8-11 in the Abstract and "It can represent the highest succession stage among the diverse range of surface-dwelling cryptogams (e.g., cyanobacteria, green algae, and lichen, which are also referred to as bio crusts) and make a major contribution to soil stability and fertility in many arid and semiarid sandy desert ecosystems (Weber et al., 2016)" in L46-49 in the Introduction.*

**Referee's comment:**

Line 214 - the chlorophyll a content of argenteum was found should be B. argenteum.

*Authors' response***:**

*We replaced "argenteum" with "B. argenteum" in L215.*

**Referee's comment:**

Table 1. "Changes in the percentage cover of a biocrust dominated by Bryum. argenteum in response to sand burial depth in spring and autumn" should be " Table 1. Changes in the percentage cover of the soil surface, a biocrust dominated by Bryum argenteum in response to sand burial depth in spring and autumn".

*Authors' response***:**

*To avoid misunderstanding, we have revised the heading of Table 1 as you suggested: "Table 1. Changes in the percentage cover of Bryum argenteum Hedw. within a biocrust in response to sand burial depth in spring and autumn.".*

**Part B: Authors' response to comments of reviewer # 2**

**Referee's comment:**

This paper presents research on the ability of biocrust mosses - in particular *Bryum argenteum* - to survive multiple stresses in dryland ecosystems. The concept is good and this research may ultimately support better land management and interventions, enabled by knowing the environmental controls on dryland biota. The novelty of the work is related to the simultaneous assessment of two stressors and their interactions: drought and burial. The experimental work appears to have been planned and carried out carefully with attention to detail, which gives me confidence in the results. The results are presented in quite a confusing way though, which made it difficult for this reader to draw own interpretation and conclusions. A few details of the method were also not clear enough for me to fully understand the results, for example I was not sure whether the moss is completely buried or whether it sticks through some of the burial treatments, which is very important. A similar problem of detail and clarity affected reading of the discussion but to a lesser extent, and the final parts of the discussion were much clearer. My suggestion for this paper is that the language and content should be revised with the aim of achieving clarity and detail relating to the objectives set out, and this may involve changing the figures too. Focusing discussion more on the fitness and adaptations of the moss is likely to help, and replacing commonly used vague terminology like "positive" and "negative" effects with specific observations or interpretations like "reduced chlorophyll content" or "increased fitness" will help further. I think that the experimental work is well conceived and of good quality, but at present it is hard to be sure whether the conclusions are fully supported by the results. Specific comments below should help the authors see examples in the manuscript relating to the above suggestions:

*Authors' response***:**

*To improve the language of our manuscript we used the services of a European editing company (International Science Editing Compuscript Ltd.). The reviewer's comments have been addressed point by point to make the manuscript be more readable and understandable. In addition, we have carefully proof-read the manuscript to minimize typographical, grammatical, and bibliographical errors. We have revised the heading of Table 1 and the legends of Figure 2-5 as suggested, and from the data shown in Table 1, it has been made clear that the*

*mosses within the biocrust were only partially buried by sand when the burial was shallow (depth ≤2 mm), while, the mosses were completely buried at a burial depth ≥ 4mm.*

*We agree that the use of specific observations or interpretations such as "reduced chlorophyll content" or "increased fitness" will help increase the clarity of the results, and have used similar phrases in the revised manuscript where appropriate. Because the major objective/focus of this study was to explore the interaction between drought and sand burial on biocrust moss, i.e., to determine if their interaction is additive or antagonistic, the terminology "positive" and "negative" effects has still been used where relevant.. We have made many modifications in to the Results section to make the text more understandable.*

**Referee's comment:**

8 "highest" is suggestive of being superior. Perhaps "latest" would be better?

45 Again use of "highest" does not seem appropriate

*Authors' response***:**

*We agree and have replaced "highest" with "latest" in L8 and L46.*

**Referee's comment:**

11 surely this is a very large niche, not small as stated

*Authors' response***:**

*We agree and have replaced "small" with "very large" in L11.*

**Referee's comment:**

58 is there a reference to support this?

*Authors' response***:**

*We have added a reference, "Li et al., 2004", to support this statement.*

**Referee's comment:**

63 is there a reference to support this (that B. argenteum is usually the pioneer)?

*Authors' response***:**

*This finding was obtained from our recent field investigation conducted throughout China's desert area, which will be presented in our next paper. To the best of our knowledge, there is no published reference in English to support this.*

**Referee's comment:**

70 "buried" perhaps a better word than "submerged"

*Authors' response***:**

*We agree and have replaced "submerged" with "buried" in L72.*

**Referee's comment:**

72-76 here setting out the importance of understanding how B. argenteum survives these multiple stressors as a main theme in the work - an interesting objective with practical applications.

*Authors' response***:**

*We agree, although we are convinced that the original sentences are sufficient to express the required meaning, and therefore we have not added any new text.*

**Referee's comment:**

85-90 it is stated that drought "protects" (benefit) and burial can "slow water loss" (benefit), so why are these then described as "mutually antagonistic" on line 87? In this context they are mutually beneficial. I think care is needed here (and earlier e.g. line 33) to note the difference between an apparently harsh environment (for humans) which is actually the niche for which certain biocrust organisms are adapted. Therefore, these harsh conditions are likely a requirement for life of the biocrust organisms being studied. Based on this one may reasonably assume that drought and burial are mutually beneficial for organisms adapted to live in this environment.

*Authors' response***:**

*Drought and sand burial are two common stressors that severely limit (even damage in most cases) the growth and distribution of biocrust moss. Here we want to show that they can also induce beneficial effects in some cases, which could potentially occur through the combination of some specific level of drought and a certain depth of sand burial.*

**Referee's comment:**

94-99 unnecessary precision of environmental parameters. The time period for which these data relate should be given.

*Authors' response***:**

*We have added "Based on meteorological records from 1956 to 2003" in L98.*

**Referee's comment:**

126 the year should be given (and on line 133)

*Authors' response***:**

*We have added "in 2013" in L131.*

**Referee's comment:**

129 "below the ground surface" - where? In situ at the extraction place in the field, or somewhere else?

*Authors' response***:**

*We have added "and transferred to Water Balance Observation Site (about 1 km from the sampling site) at Shapotou Desert Research and Experiment Station, Chinese Academy of Sciences." in L128-129.*

**Referee's comment:**

138 please explain the burial a bit more, and refer to this in the introduction and discussion too as appropriate. It is necessary to know whether the burial completely covers the moss, or whether it sticks up through the added sand. This has major implications for the interpretation and understanding of the work.

*Authors' response***:**

*We believe that burial phenomena can be easily and clearly envisaged from the data shown in Table 1 (similar to Jia et al., Soil Biol. Biochem., 40, 2827-2834, 2008; Mart ńez, Plant Ecol., 145, 209-219, 1999; Maun, M. A. Can. J. Bot., 7, 713–738, 1998; Maun, M. A. Coastal Dunes: Ecology and Management, Springer, 119–136, 2008.). The mosses within the biocrust were only partially buried by sand when the depth of burial was shallow (depth ≤2 mm), wheeas, the mosses were completely buried when the burial depth≥ 4mm. To avoid any*

*misunderstanding, we have revised the heading of Table 1 to "Table 1. Changes in the percentage cover of Bryum argenteum Hedw. within a biocrust in response to sand burial depth in spring and autumn.".*

**Referee's comment:**

143 The experiment duration seems rather short but nothing can be done about that now. Perhaps the duration can be explained / justified?

*Authors' response***:**

*Yes, 72 d is a short duration for each season. We did our best to ensure that the duration of the experiment was as close to the actual duration of each season, and to keep the duration identical in both spring and autumn, but we are convinced that the duration attained was sufficient to justify the conclusions of the study.*

**Referee's comment:**

147 Would deposited sand be naturally blown off in the field? The answer to this is of interest in relation to how the moss adapts to burial. If the deposit is never blown off then the moss needs to abandon the buried chlorophyll and invest in the tip, however if it might be blown off then it would make sense to retain the buried chlorophyll for a while in-case it will be exposed later.

*Authors' response***:**

*Yes, sand deposited above the biocrust moss can be blown off under field conditions. We have discussed this in L357-358 in the Discussion section.*

**Referee's comment:**

152 A nice idea to minimise the edge effect.

159 Not clear if "shoot elongation" is the same as "shoot upgrowth" on line 157 – if same then please use same terminology, if different then please explain.

*Authors' response***:**

*For biocrust moss under sand burial, according to our observations, "shoot elongation" and "shoot upgrowth" have the same meaning.*

**Referee's comment:**

166 Again nice attention to detail in the experimental method here (randomising positions)

185 I think the unit is wrong here

*Authors' response***:**

*We have replaced "g" with "*rpm*" in L187-188.*

**Referee's comment:**

211 and throughout results section: It is not clear what negative and positive effects are. Instead, just state which measurements changed and how they changed (e.g. decrease or increase in chlorphyll content). In general I found the results section quite difficult to follow because of this. Furthermore in some cases the language used unclear phrases (e.g. "decrease in autumn being significantly lower" line 206). There is some interpretation in results which should instead be in discussion.

*Authors' response***:**

*To make the text clearer, we revised the original sentences to "Drought uniformly imposed negative effects (the*

*slopes of the fitted lines were negative) on the chlorophyll a content (Fig. 2a, b), whereas burial by sand had a dual effect (the slopes of the fitted lines exhibited both positive and negative values) on the chlorophyll a content (Fig. 2c, d)."in L215-217,"Drought consistently exerted negative effects (the slopes of the fitted lines were negative) on the PSII photochemical efficiency (Fig. 3a, b), while burial by sand had a dual effect (the slopes of the fitted lines showed both positive and negative values) on the PSII photochemical efficiency (Fig. 3c, d)."in L229-232, "Drought imposed negative effects (the slopes of the fitted lines were negative) on the regeneration potential of detached shoots of B. argenteum (Fig. 4a), while burial by sand had a dual effect (the slopes of the fitted lines displayed both positive and negative values) on the regeneration potential (Fig. 4b)." in L242-244 and "Although B. argenteum shoots were generally less elongated in spring than that under the same treatment in autumn, drought and sand burial, according to the slope values of the corresponding fitted lines, had both negative and dual effects on shoot elongation, respectively" in L252-253.*

*In addition, we replaced "decrease" with "amplitude decrease" in L209.*

**Referee's comment:**

284-286 I'm not sure if the effects observed have explained the moss distribution as claimed. Perhaps more explaination of this needed, or remove.

*Authors' response***:**

*To be more accurate, we replaced "pattern of" with "survival and" in L293.*

**Referee's comment:**

288 The discussion here is interesting, effectively summarising findings for the most part but still a bit confusing in relation to what is a positive or negative, which seems to partly contradict the introductory section e.g. lines 85-90

*Authors' response***:**

*We think that there is no contradiction and any confusioncan be explained as follows. The few studies referred to in the Introduction section showed that drought and sand burial can separately impose only one-way, beneficial effects on a biocrust moss, as well as intuitively causing harm. While, this study showed that, drought and sand burial on their own had more complex, dual effects in most cases, the beneficial and detrimental effects caused by either factor alone can emerge simultaneously and vary with the changes in the severity of drought or depth of burial. The combination of these factors indisputably leads to more complicated effects on biocrust moss. We are convinced that we have clearly indicated the single or interactive influences on B. argenteum for every parameter tested.*

**Referee's comment:**

370-374 this is a clear and useful outcome of the work. Probably best not to mention the un-published results though.

*Authors' response***:**

*We have deleted the sentence with the un-published results.*

**Referee's comment:**

380 This section also presenting clear and useful findings

Table 1. The precision to 3dp seems excessive, making it less clear

*Authors' response*:

*We have redrawn Table 1 to make it clearer.*

**Referee's comment:**

Figures 2-5. These are quite complicated and can't be fully understood based on the legend. For instance what is Control, 0.5mm, 1mm etc written at the top? What are the units or scale of drought severity? These details may be elsewhere but it should be possible to interpret the figures alone. I think some work is needed to make these a bit clearer, or if not possible consider using them as supplementary figures and replace with something less detailed which is easier to interpret.

*Authors' response*:

*To improve the clarity and ease of understandability for reader, we have revised the legends of Figures 2-5, as shown below:*

*Figure 2. Changes in the chlorophyll a content of the biocrust moss Bryum argenteum Hedw. following exposure to combinations of single (receiving the natural precipitation amount, control), double (receiving 1/2 of the natural precipitation amount) and fourfold (receiving 1/4 of the natural precipitation amount) drought and 0 (control), 0.5, 1, 2, 4, and 10 mm depths of sand burial in spring (a, c) and autumn (b, d). Bars represent means (± SE). Different letters indicate significant differences between different drought severities and sand burial depth treatments at the p <0.05 level, as determined using a least significant difference (LSD) post-hoc test.*

*Figure 3. Changes in the PSII photochemical efficiency (Fv/Fm) of the biocrust moss* Bryum argenteum *Hedw. following exposure to combinations of single (receiving the natural precipitation amount, control), double (receiving 1/2 of the natural precipitation amount) and fourfold (receiving 1/4 of the natural precipitation amount) drought and 0 (control), 0.5, 1, 2, 4, and 10 mm depths of sand burial in spring (a, c) and autumn (b, d). Bars represent means (± SE). Different letters indicate significant differences between different drought severities and sand burial depth treatments at the* p *<0.05 level as determined using a least significant difference (LSD) post-hoc test in spring and autumn.*

*Figure 4. Changes in the protonemal area of detached shoots of biocrust moss* Bryum argenteum *Hedw. following exposure to combinations of single (receiving the natural precipitation amount, control), double (receiving 1/2 of the natural precipitation amount) and fourfold (receiving 1/4 of the natural precipitation amount) drought and 0 (control), 0.5, 1, 2, 4, and 10 mm depths of sand burial in spring. Bars represent means (± SE). Different letters indicate significant differences between different drought severities and sand burial depth treatments at the* p *<0.05 level as determined using a least significant difference (LSD) post-hoc test.*

*Figure 5. Changes in the maximal shoot elongation of biocrust moss* Bryum argenteum *Hedw. following exposure to combinations of single (receiving the natural precipitation amount, control), double (receiving 1/2 of the natural precipitation amount) and fourfold (receiving 1/4 of the natural precipitation amount) drought and 0 (control), 0.5, 1, 2, 4, and 10 mm depths of sand burial in spring (a, c) and autumn (b, d). Bars represent means (± SE). Different letters indicate significant differences between different drought severities and sand*

*burial depth treatments at the* p *<0.05 level as determined using a least significant difference (LSD) post-hoc test.*

*In addition, Figure 4 was replaced with the correct figure.*

**Part C: Authors' response to comments of Associate Editor**

**Referee's comment:**

As a title of your manuscript, I would suggest "Antagonistic effects of drought and sand burial enable survival of biocrust moss Bryum argenteum in an arid sandy desert"

***Authors' response*:**

*We agree and have revised the title of our manuscript to "Antagonistic effects of drought and sand burial enable survival of biocrust moss Bryum argenteum in an arid sandy desert".*

**A list of all relevant changes made in the manuscript**

**1.Title:** changed to "Antagonistic effects of drought and sand burial enable survival of biocrust moss *Bryum argenteum* in an arid sandy desert".

**2. Key words:** Added "antagonistic effect, drought, sand burial, *Bryum argenteum*".

**3. Authors and affiliations:** We would like to add Yixuan Li as the last author in the authorship list, as suggested by Rongliang Jia and approved by other co-authors, considering Li's contributions to our experiment, data processing, manuscript writing and revising.

**4. Abstract section:**

1) We have replaced the old sentence to "It can represent the latest succession stage among the diverse range of surface-dwelling cryptogams (e.g., cyanobacteria, green algae, and lichen, which are also referred to as biocrusts) and it can make a major contribution to soil stability and fertility in many arid sandy desert ecosystems" in L8-11.

2) We have replaced "small" with "very large" in L11.

**5. Introduction section:**

1) We have replaced "increases" with "increase" in L34.

2) We have replaced the old sentence to "It can represent the latest succession stage among the diverse range of surface-dwelling cryptogams (e.g., cyanobacteria, green algae, and lichen, which are also referred to as biocrusts) and make a major contribution to soil stability and fertility in many arid and semiarid sandy desert ecosystems" in L45-48..

3) We have added a reference, "Li et al., 2004" L56.

4) We agree and have replaced "submerged" with "buried" in L70.

**6. Materials and methods section:**

1) We have added "Based on meteorological records from 1956 to 2003" in L96.

2) We have added "in 2013" in L129.

3) We have added "and transferred to Water Balance Observation Site (about 1 km from the sampling site) at Shapotou Desert Research and Experiment Station, Chinese Academy of Sciences." in L126-127.

4) We have replaced "g" with "rpm" in L185-186.

**7. Results section:**

1) We have added "amplitude decrease" in L207.

2) We have revised the original sentences to "Drought uniformly imposed negative effects (the slopes of the fitted lines were negative) on the chlorophyll a content (Fig. 2a, b), whereas burial by sand had a dual effect (the slopes of the fitted lines exhibited both positive and negative values) on the chlorophyll a content (Fig. 2c, d)." in L213-215.

3) We have revised the original sentences to "Drought consistently exerted negative effects (the slopes of the fitted lines were negative) on the PSII photochemical efficiency (Fig. 3a, b), while burial by sand had a dual effect (the slopes of the fitted lines showed both positive and negative values) on the PSII photochemical

efficiency (Fig. 3c, d)."in L227-230.

4) We have revised the original sentences to "Drought imposed negative effects (the slopes of the fitted lines were negative) on the regeneration potential of detached shoots of *B. argenteum* (Fig. 4a), while burial by sand had a dual effect (the slopes of the fitted lines displayed both positive and negative values) on the regeneration potential (Fig. 4b)." in L240-242

5) We have revised the original sentences to "Although *B. argenteum* shoots were generally less elongated in spring than that under the same treatment in autumn, drought and sand burial, according to the slope values of the corresponding fitted lines, had both negative and dual effects on shoot elongation, respectively" in L251-253.

**8. Discussion section:**

1) We have replaced "*B. argenteum* Hedw. usually" with "*Bryum argenteum* Hedw. generally".

2) To be more accurate, we have replaced "pattern of" with "survival and" in L291.

3) We have deleted the sentence "The results of our latest pilot experiment (unpublished) support this proposition." before "The use of such a technique is also in agreement with Maestre et al. (2006)" in L380.

**9. Acknowledgments section:** We have added "We gratefully acknowledge Bettina Weber, David Elliott and one anonymous reviewer for their invaluable comments on the manuscript." in L398-399.

**10. References section:** We have replaced "outer" with "Outer" in L437.

**11. Table 1:** The heading of Table 1 was revised to avoid misunderstanding, and Table 1 was redrawn to make it clearer.

**12. Figures:**

1) The legends of Figures 2-5 were revised to improve the clarity and ease of understandability.

2) Figure 4 was replaced with the correct figure.

[revised manuscript text omitted]

---

## Author Response (AR2)

**Authors' response to Associate Editor's comment**

**Associate Editor's comment:**

meanwhile I checked your manuscript again and saw that you responded accurately upon the reviewer comments. However, figures 2-5 could be improved considerably if you use 3-D-plots to show the response of the mosses upon 2 different parameters. Doing this, the results can be interpreted more easily and plots will be reduced to half. Thus, please change these figures accordingly and submit your manuscript again.

**Authors' response:**

*We agree and Figures 2-5 have been redrawn to 3-D-plots by using MATLAB software, and the legends of figures 2-5, Results section and Statistics in Materials and methods section were revised accordingly.*

**A list of all relevant changes made in the manuscript**

**1. Statistics in Materials and methods section:**

1) added one paragraph: "3-D-plots were produced using MatlabR2014a (The MathWorks Inc., Natick, MA, USA) to show the responses of chlorophyll a content, PSII photochemical efficiency, regeneration potential, and shoot upgrowth of the biocrust moss *B. argenteum* to a combination of three levels of drought severity and six depths of sand burial." in L190-192.

2) deleted one paragraph: "Linear regressions were performed in the ORIGIN 8.5 (OriginLab, Northampton, USA) to depict any trend in the chlorophyll a content, PSII photochemical efficiency, regeneration potential of detached moss shoots, and maximal shoot elongation to increases in drought severity /burial depth."

**2. Results section:**

1) We have deleted "(the slopes of the fitted lines were negative)", "(Figure 2a,b)" and "(the slopes of the fitted lines exhibited both positive and negative values)" in L213-214.

2) Replaced "Fig. 2c, d" in L214, L218 and "Fig. 2a, b" in L219 with "Fig. 2".

3) We have deleted "(the slopes of the fitted lines were negative)", "(Figure 3a,b)" and "(the slopes of the fitted lines showed both positive and negative values)" in L224-225.

4) Replaced "Fig. 3c, d" in L225, L230 and "Fig. 3a, b" in L231 with "Fig. 3".

5) We have deleted "(the slopes of the fitted lines were negative)", "(Figure 4a)" and "(the slopes of the fitted lines displayed both positive and negative values)" in L235-236.

6) Replaced "Fig. 4b" in L236 with "Fig. 4" and added "Fig. 4" in L241 and L243.

7) We have deleted ", according to the slope values of the corresponding fitted lines," and "(Fig. 5c, d)" in L247-250.

8) Replaced "Fig. 5a, b" with "Fig. 5" in L251.

**3. Figures:**
Figures 2-5 have been redrawn to 3-D-plots by using MATLAB software and the legends of Figures 2-5 were revised accordingly.

**4. others:**
replaced "," with ";" in L51.

[revised manuscript text omitted]

---

## Author Response (AR3)

**Authors' response to Associate Editor's comment**

**Associate Editor's comment:**

Dear Dr. Jia, thank you very much for preparing the 3-D plots.
Minor technical issues still caught my eye.

1. In Fig. 1, it is not clear which letter belongs to which sub-figure; please rearrange the letters to make that clearer. In the legend please explain the letters in the right order; otherwise it is confusing.
2. In Fig. 2 b, please make sure that you don't have 0.5 intervals on the x-axis.
3. In Fig. 2-5 the explanation of the x-axis is not clear; as an example, here is a possible legend for figure 2; please adapt the other legends accordingly:

"Changes of the chlorophyll a content of the biocrust moss Bryum argenteum Hedw. following exposure to natural precipitation (control; 1), half of the natural precipitation amount (2), one fourth of the natural precipitation amount (4), combined with 0 (control), 0.5, 1, 2, 4, and 10 mm depth of sand burial in spring (a) and autumn (b). Symbols represent means +- SE. ..."

Kind regards,

Bettina Weber

*Authors' response:*

Dear Bettina,

We very much appreciate the careful reading of our manuscript and professional comments to our manuscript. We have revised the manuscript point by point according to your advices, which thoroughly improved the quality of our manuscript, as the following:
1. We have rearranged the letters in Fig. 1 and explained them in the right order in the legend.
2. We have redrawn Fig. 2 b and deleted 0.5 intervals on the x-axis; we also have redrawn Fig. 5 b and added 10 on the y-axis.
3. We have revised the legends of Fig. 2-5 as you suggested.

We hope these changes would strengthen the rationale for publication in BG.

Our best wishes,,

Rongliang Jia

Yun Zhao

Yanhong Gao

Rong Hui

Haotian Yang

Zenru Wang

Yixuan Li

**A list of all relevant changes made in the manuscript**

**1. Fig. 1:**

1) Rearranged the locations of the letters.

2) Changed the legend to: "Location and main landscapes of the Shapotou zone in the southeastern edge of the Tengger Desert. As a pioneer species, *Bryum argenteum* Hedw. has colonized and flourishes on the soil surface of the revegetation area (a) by controlling burial stress through the combined application of wind barriers, straw checkerboards, and the planting of shrubs without irrigation (b) in the previously shifting sand dunes (c) 60 years ago.".

**2. Fig. 2:** we have redrawn Fig. 2 b and deleted 0.5 intervals on the x-axis.

**3. Fig. 5:** we have redrawn Fig. 5 b and added 10 on the y-axis.

**4. Legends of Fig. 2-5 have been revised, as the following:**

[revised manuscript text omitted]

---

## Author Response (AR4)

**Authors' response to Associate Editor's comment**

**Associate Editor's comment:**

Dear authors,

I think now your manuscript is ready for publication. Can you just provide an update of the legend of Figure 1. I would suggest to formulate it in the following way: "Figure 1: Location and main landscapes of the Shaputou region at the southeastern edge of the Tengger Desert. As a pioneer species ...(a); this was achieved by controlling burial stress through ...(b); 60 years ago, the area was characterized by shifting sand dunes (c)."

Kind regards,

Bettina Weber

***Authors' response*:**

Dear Bettina,

Great! We have revised the legend of Fig. 1 according to your suggestion.
Thank you for your kind help. We should express our great appreciation for your professional assistance.

Our best wishes,,

[revised manuscript text omitted]